The determination of thiocyanate in the blood plasma and holding water of Amphiprion clarkii after exposure to cyanide

Bonanno J. Alexander 1 2
Breen Nancy E. 3
Tlusty Michael F. 1
Andrade Lawrence 4
Rhyne Andrew L. arhyne@rwu.edu 5
1 School for the Environment, University of Massachusetts at Boston , Boston , MA , United States of America
2 Current affiliation: Takara Bio USA, Inc. , San Jose , CA , United States of America
3 Department of Chemistry, Roger Williams University , Bristol , RI , United States of America
4 Dominion Diagnostics , North Kingstown , RI , United States of America
5 Department of Biology, Marine Biology, and Environmental Science, Roger Williams University , Bristol , RI , United States of America
Pawlik Joseph
Electronic publication date: 2021 Dec 7
Publication date: 2021
Volume: 9
Electronic Location ID: e12409
Received 2020 May 7; Accepted 2021 Oct 8
Copyright: ©2021 Bonanno et al.
Copyright year: 2021
Copyright holder: Bonanno et al.
License: This is an open access article distributed under the terms of the Creative Commons Attribution License, which permits unrestricted use, distribution, reproduction and adaptation in any medium and for any purpose provided that it is properly attributed. For attribution, the original author(s), title, publication source (PeerJ) and either DOI or URL of the article must be cited.
License URL: https://creativecommons.org/licenses/by/4.0/

Keywords: Cyanide Fishing, Aquarium Trade, Thiocyanate, Wildlife Trade, IUU Fishing

Funding: The Pet Industry Joint Advisory Council This work was supported by the Pet Industry Joint Advisory Council. The funders had no role in study design, data collection and analysis, decision to publish, or preparation of the manuscript.

==============================
The illegal practice of cyanide fishing continues throughout the Indo-Pacific. To combat this destructive fishing method, a reliable test to detect whether a fish has been captured using cyanide (CN) is needed. We report on the toxicokinetics of acute, pulsed CN exposure and chronic thiocyanate (SCN) exposure, the major metabolite of CN, in the clownfish species, Amphiprion clarkii. Fish were pulse exposed to 50 ppm CN for 20 or 45 s or chronically exposed to 100 ppm SCN for 12 days and blood plasma levels of SCN were measured. SCN blood plasma levels reached a maximum concentration (301–468 ppb) 0.13–0.17 days after exposure to CN and had a 0.1 to 1.2 day half-life. The half-life of blood plasma SCN after chronic exposure to SCN was found to be 0.13 days. Interestingly, we observed that when a fish, with no previous CN or SCN exposure, was placed in holding water spiked to 20 ppb SCN, there was a steady decrease in the SCN concentration in the holding water until it could no longer be detected at 24 hrs. Under chronic exposure conditions (100 ppm, 12 days), trace levels of SCN (∼40 ppb) were detected in the holding water during depuration but decreased to below detection within the first 24 hrs. Our holding water experiments demonstrate that low levels of SCN in the holding water of A. clarkii will not persist, but rather will quickly and steadily decrease to below detection limits refuting several publications. After CN exposure, A. clarkii exhibits a classic two compartment model where SCN is eliminated from the blood plasma and is likely distributed throughout the body. Similar studies of other species must be examined to continue to develop our understanding of CN metabolism in marine fish before a reliable cyanide detection test can be developed.

Introduction

Coral reef ecosystems boast the greatest diversity of marine life globally (Bruno & Selig, 2007; Reaka-Kudla, 1997) making them one of the most valuable ecosystems on our planet, yet they are also one of the most threatened (Descombes et al., 2015; Hughes et al., 2017). These ecosystems are being stressed to their tipping point largely by climate change, but other anthropogenic activities are playing a role in their destruction including illegal fishing practices (Burke, Reytar & Spalding, 2012; Dietzel et al., 2020; Hoegh-Guldberg, 1999; Hughes et al., 2017).

One widely used illegal fishing practice in the Indo-Pacific that threatens coral reefs is the use of cyanide (CN) to capture reef fish. Cyanide has been used as a stunning agent to collect fish for over 60 years (Bellwood, 1981; Frey, 2013; Lewis & Tarrant Jr, 1960; Rubec et al., 2003). This practice is used for the marine aquarium trade (MAT) and the live reef food fish trade (LRFFT) in the Indo-Pacific region where 75% of all coral reefs are located (Barber & Pratt, 1997; Bruno & Selig, 2007; Davis, Murray & Katsiadaki, 2017; Graham, 2001; Losada & Bersuder, 2017; Rubec, 1986). Cyanide fishing involves dissolving tablets of potassium or sodium cyanide in a squirt bottle filled with seawater. Fisherfolk then squirt the concentrated CN solution onto the fish that inhabit the reef and in crevices in the reef where the fish hide. At sublethal doses, which vary depending on the size and species of the fish, CN temporarily paralyzes the fish, immobilizing them for easy capture. Despite being illegal, this fishing practice continues to be used throughout the region. Fisherfolk rely on middlemen to provide cyanide, food, boats, and receive minimal pay in return for high volumes of fish caught with cyanide (Rubec et al., 2001). Policing against this fishing method is difficult, as there is no easy and definitive way to test if a fish has been captured using CN. As a result, enforcement of CN fishing laws is challenging, if not impossible (Losada & Bersuder, 2017).

The squirt bottle method of CN delivery makes it difficult for fishers to control the amount of solution dispensed, resulting in widely variable doses received. As CN is nonselective, and dosing is not precise, this method often kills both the targeted species and non-targeted species in the squirt zone (Frey, 2013). Even when fish survive the exposure, there are reports of increased fish mortality post-capture associated with this fishing practice (Cervino et al., 2003; Davis, Murray & Katsiadaki, 2017; Hall & Bellwood, 1995; Pyle, 1993; Rubec, 1986; Rubec et al., 2003). Furthermore, stony corals, which are an important structural component of coral reef ecosystems, are often damaged from exposure to CN during the fishing process (Cervino et al., 2003; Jones & Hoegh-Guldberg, 1999). Once the fish are stunned, fisherfolk also may break stony corals to gain access to the fish, increasing the long-term damage to reefs (Bruckner & Roberts, 2008).

Attempts to curtail the use of CN as a capture method via post-capture testing have not been successful (Dalabajan, 2005; Erdmann, 1999). The Cyanide Detection Test (CDT), which uses a CN ion-selective electrode to measure CN found in tissues of fish, was developed by the International Marinelife Alliance (IMA) and the Philippines Bureau of Fisheries and Aquatic Resources (BFAR) and was used in the Philippines in the 1990s (Barber & Pratt, 1997; Manipula, Suplido & Astillero, 2001). The CDT test was developed by spiking samples of fish homogenate with CN, rather than systematically exposing marine fish to cyanide and testing the exposed fish. There is no reported validation of this test for CN on live fish with known CN exposure. It was suggested that the test was unreliable as an indicator of CN exposure primarily due to interferences resulting in false positives (Balboa, 2017; Mak, Yanase & Renneberg, 2005). Bruckner & Roberts (2008) thoroughly documented the shortcomings of the IMA CDT. The test required lethal sampling, was time-consuming, labor-intensive, and was never properly field-tested and verified. Mak, Yanase & Renneberg (2005) attempted to validate the IMA CDT on fish exposed to CN in their laboratory. They exposed fish to enough CN to induce mortality, but found that CN was not detected in the homogenates of the exposed fish using the CDT methodology (Mak, Yanase & Renneberg, 2005). Other CN detection methods have been proposed (Mak, Yanase & Renneberg, 2005), but the presumed rapid detoxification of CN is the major difficulty in detecting CN exposure. If marine fish have a similar detoxification pathway to mammals, and it is believed they do because of the presence of the enzyme rhodanese, any test for CN in fish must be administered very soon after exposure, likely within hours (Day et al., 2018; Logue et al., 2010).

As an alternative to detecting CN directly, thiocyanate (SCN), a metabolite of CN exposure, is often used as an indicator of CN exposure (Breen et al., 2019; Day et al., 2018; Youso, Rockwood & Logue, 2012). In mammals, the CN detoxification pathway is well established. Two sulfur-transferases, rhodanese, and 3-mercaptopyruvate sulfurtransferase are responsible for catalyzing the formation of SCN from CN by sulfuration (Day et al., 2018), and SCN is eliminated in urine by the kidneys. In marine fish, the method of elimination has not yet been determined, but Vaz et al. (2012) and Rubec et al. (2003) assumed a similar pathway to mammals because of the presence of the rhodanase enzyme. In a now refuted study, Vaz et al. (2012) claimed that SCN excreted in the urine of the marine fish Amphiprion clarkii exposed to CN could be detected in their holding water for at least 28 days. The same group used this technique to monitor the MAT in the EU for the presence of cyanide fishing in the exporting countries (Vaz, Esteves & Calado, 2017). The hope was that this technique could serve as a test for CN exposure in marine fish, but in the nine years that have passed since it was first reported, it has never been replicated by any other lab even though multiple laboratories have attempted to do so (Breen et al., 2018; Herz et al., 2016; reviewed by Murray et al., 2020). Breen et al. (2018) demonstrated by using a mass balance calculation that the SCN concentrations reported by Vaz et al. (2012) and the concomitant dose of CN that the fish would have had to receive given the reported SCN concentrations were an order of magnitude higher than all known LD50s for CN in vertebrate species. If marine fish excrete SCN into the water after exposure to CN, then the concentration of SCN in the holding water must be well below current detection limits.

The next logical step in developing our understanding of cyanide detoxification in marine fish is to look for SCN or CN directly in bodily fluids of marine fish exposed to CN. This will provide much needed information on cyanide toxicokinetics in marine fish and thus eliminate the need for speculation and uncertainty when applying mammalian models to marine fish. Indeed, elevated levels of SCN in the blood plasma of marine fish after acute pulsed exposure to CN have been recently reported (Breen et al., 2019). SCN concentrations in the blood plasma of the laboratory cultured marine fish Amphiprion ocellaris were observed to be above control levels up to 41 days post-exposure to CN, which was the longest time sampled. This study reported both a fast and a slow half life corresponding to an initial fast elimination of SCN in blood plasma followed by a much slower elimination from the blood plasma as evidenced by long residence times of low levels of SCN above that found in the controls. Breen et al. (2019) speculated that the observation of both a fast and slow elimination rate of SCN from the blood plasma might be due to multiple elimination pathways in marine fish. These rates could also be governed by the availability of sulfur donors and the rate of diffusion from organs and tissues with limited blood flow (Day et al., 2018).

Breen et al. (2019) were the first to report the half-life of SCN in the blood plasma of a marine fish following CN exposure. The report was for a single species from an aquacultured stock with limited genetic diversity within the replicate fish. Over 2,300 documented fish taxa of various sizes are traded in the MAT (Rhyne et al., 2012; Rhyne et al., 2017) and while the A. ocellaris work is a vital first step in understanding the fate of CN in marine fish, the species and size dependent variation of CN metabolism is not known. This work must be extended to other species and the endogenous levels of SCN in marine fish in the wild must also be determined before any test that relies on SCN as a marker for CN exposure can be considered for widespread use.

In following up our work on A. ocellaris (Breen et al., 2019), this study examines the toxicokinetics of a congeneric clownfish species (A. clarkii), the same species used by Vaz et al. (2012). In an acute exposure study, A. clarkii were pulse exposed to CN and the rate of elimination of SCN from the blood plasma following exposure was determined by ultra high-performance liquid chromatography (UHPLC) with mass spectrometry (MS). In a chronic exposure study, A. clarkii were exposed to SCN and the rate of elimination of SCN from the blood plasma was determined using high performance liquid chromatography (HPLC) with UV absorbance detection. The toxicokinetics of SCN elimination is reported for both exposures. The holding water containing the A. clarkii that were chronically exposed to SCN was tested for the presence of SCN during depuration. With chronic, high-level exposure, the possibility of detection of SCN in holding water during depuration via bodily fluid excretion is drastically enhanced.

Methods

Test species

Experiments were approved by the Roger Williams University Institutional Animal Use and Care Committee (Approval #R180820). A. clarkii of approximately 6–12 months of age were cultured in captivity at Roger Williams University or Sea & Reef Aquaculture, Franklin, ME, thereby ensuring they were not collected with or previously exposed to CN (Table 1). For all experiments, the temperature of the water holding the fish was maintained to 25 °C by placing the holding tank containing the fish in either a temperature-controlled room or in a warm water bath. All fish were fed pelletized food (Skretting Green Granule one mm) once per day unless otherwise noted. Water quality was maintained through daily water changes (100%) at a salinity of 30 with light aeration. Natural seawater was used to house fish throughout each experiment. All seawater was mechanically filtered down to one micron, chlorinated, and then dechlorinated with UV sterilization and carbon filtration and routinely tested for the presence of SCN.

Table 1 Average weight, sample size (n), and controls of A. clarkii pulse exposed to an acute dosage of 50 ppm CN for either 20 or 45 s across 2 trials and 2 chronic exposures of 100 ppm SCN for 12 days.

There were two groups (small/large) of A. clarkii used for CN 2 trial however fish size did not affect SCN half-life.

Trial	Exposure time	Dose (ppm)	Sampling duration (days)	n	Controls	Average weight (grams ± S.D.)	
CN 1	20 s	50	13	44	5	3.1 ± 0.6	
CN 1	45s	50	13	47	3.6 ± 0.8	
Additional Controls	9	18.9 ± 5.7	
CN 2	45 s	50	72	41	4	12.8 ± 9.4 Small: 5.1 ± 1.3 Large: 21.0 ± 6.8	
SCN 1	12 d	100	16	37	4	6.1 ± 2.7	
SCN 2	12 d	100	2	10	3	1.9 ± 0.2	

CN and SCN exposures

During the acute CN treatments, fish were pulse exposed to a solution of 50 ppm CN (NaCN, MilliporeSigma, St Louis, MO, USA 380970) for 20 s and 45 s in the first trial and 45 s in the second trial (Table 1). Groups of fish were placed in baskets and immersed in the CN solution for the pre-determined time. The fish were then rinsed by transferring the basket containing the fish to two successive seawater baths from the same source and at a salinity of 30. After rinsing, fish were housed by their exposure group in round, 20 L polycarbonate tanks containing seawater with light aeration. The first CN exposure trial examined the SCN concentration in blood plasma of 4–7 fish for each sampling point during the first 13 days post-exposure. Collection times were approximately 1, 3, 6, 8, and 15 hrs and 1, 2, 3, 7, 12, and 13 days post-exposure. For the second CN exposure trial, the SCN concentration in the blood plasma of 4–8 exposed fish was monitored until two months post-exposure. Collection times were 4 hrs, 12 hrs, 2, 7, 18, 50, and 72 days post-exposure. Blood plasma was also collected from a total of 18 control fish not exposed to CN across the two trials.

For the SCN treatment, fish were chronically exposed to 100 ppm SCN (NaSCN, MilliporeSigma, St Louis, MO, USA 467871) for 12 days (Table 1). Fish were housed in three round, 20 L polycarbonate tanks holding 15 L of seawater with 12–13 fish in each tank. Fish were fed daily and complete water changes were performed after each feeding. After 12 days, each fish was rinsed thoroughly by submersing it in three consecutive seawater baths to remove all SCN from the surface of the fish. After rinsing, the fish were housed together with their respective exposure groups in three separate 20 L polycarbonate tanks for depuration. During depuration, the tanks and fish were maintained as above. Once depuration was initiated, 4 - 9 fish were sampled at 1, 2, 4, 8, and 16, hrs and 1, 3, and 16 days. While depurating, the holding water of the fish was also sampled over the first 72 hrs to test for the presence of excreted SCN. Four control fish were also sampled at the end of the depuration period.

SCN in holding water

A second SCN chronic exposure trial was carried out on A. clarkii with an even more thorough rinsing procedure. Fish (n = 10) were exposed to 100 ppm SCN for 12 days in a 20 L polycarbonate tank as described above. To begin the depuration period, individual fish were rinsed by submersion in three different seawater baths (4 L) and placed in covered beakers containing 500 mL of seawater, one fish per beaker. The rinse water was changed after each fish was rinsed. To monitor for the possibility of cross contamination, an additional 6 beakers were held in the same area, 3 containing non-exposed fish, and 3 with just seawater, no fish. Water samples (one mL) were collected from each beaker before the addition of any fish, and then immediately after the addition of the fish. Following this, samples were collected at 2, 4, 8, 12, 24, 36, 48, and 72 hrs. Water changes were performed every 24 hrs. Sampling at 24, 36, 48, and 72 hrs was carried out just prior to the water change. Fish were fed during exposure but not during depuration. The SCN blood plasma levels of half the fish were measured at 48 hrs and the SCN blood plasma levels for the fish remaining were measured after the final sampling at 72 hrs when the study was concluded.

An additional holding water experiment was undertaken to further examine the previous results. Here, fish (n = 10) with no known previous exposure to SCN, were placed in individual 500 mL beakers of seawater spiked to 20 ppb with SCN. Water samples were collected before spiking the water, after spiking the water with SCN (t = 0), and at 2, 4, 8, and 16 hrs. At 24 hrs the water was sampled, changed, spiked back to 20 ppb, and re-sampled. After a second 24 hr interval (48 hrs total), the water was sampled. The SCN blood plasma levels of half the fish were measured at 24 hrs before the water change and re-spike. The SCN blood plasma levels for the fish remaining were measured after the final sampling at 48 hrs when the study was concluded. SCN blood plasma was measured as described below. In both experiments holding water samples were collected before adding fish or SCN to test the water for the presence of SCN.

Blood plasma collection

Blood plasma was collected for SCN analysis following the method described by Breen et al. (2019). The exact dates and times are noted in the Supplemental Information. Fish were heavily anesthetized with tricaine methanesulfonate (Western Chemical Inc., Ferndale, WA, USA) at a concentration of 200 ppm, buffered 2:1 with sodium bicarbonate in saltwater, and then dried and weighed on an analytical balance. Blood was collected by severing of the caudal peduncle with a #21 surgical blade. For fish weighing less than 4 g, blood was collected in 40 mm heparinized microhematocrit tubes (Jorvet, Loveland, CO, USA) while for fish weighing more than 4 g, blood was collected in 125 µL heparinized microcapillary blood collection tubes (RAM Scientific, Nashville, TN, USA). Following blood collection, fish were euthanized via pithing. The 40 mm heparinized microhematocrit tubes were then centrifuged (ZipCombo Centrifuge, LW Scientific, Lawrenceville, GA, USA) at room temperature at 3,000 rpm (470 g) for two min followed by 6,000 rpm (1,900 g) for five min to separate the red blood cells from the blood plasma. The tubes were then snapped at the plasma and red blood cell interface and the blood plasma was aspirated from the capillary tubes into pre-weighed 1.7 mL centrifuge tubes and then re-weighed to determine the mass of blood plasma collected. The 125 µL micro-capillary blood collection tubes were spun at 12,000 rpm (6,900 g) for 12 min at room temperature. The top blood plasma layer was then pipetted into a pre-weighed 1.7 mL centrifuge tube. The centrifuge tubes containing the blood plasma were then reweighed to determine the blood plasma mass. Blood plasma was stored in these centrifuge tubes at −80 °C until analyzed.

Thiocyanate analysis

Before blood plasma analysis, proteins were precipitated with cold HPLC grade acetonitrile (MilliporeSigma, St Louis, MO, USA) (Blanchard, 1981). Acetonitrile was added in the ratio of 1:5 (v/v) and the solution was vortexed for 20 s and then centrifuged at 12,000 rpm (6,900 g) for 10 min at room temperature. The supernatant was transferred to a new 1.7 mL centrifuge tube and the acetonitrile was evaporated with nitrogen gas in an Zymark TurboVap LV Evaporator (Zymark, Hopkinton, MA, USA) at 70 °C. The residue was reconstituted in enough HPLC grade water (MilliporeSigma, St Louis, MO, USA) to make a 1:5 or 1:10 dilution of the plasma collected and vortexed for 20 s.

For the first chronic exposure to 100 ppm SCN experiment, the SCN in the blood plasma was analyzed using the HPLC-UV method described by Breen et al. (2019) and Breen et al. (2018) originally adopted from Rong, Lim & Takeuchi (2005). For the SCN blood plasma half-life experiments following CN exposure, the second SCN chronic exposure experiment (with holding water sampling), and the subsequent holding water experiments, SCN concentrations were analyzed following the method developed by Bhandari et al. (2014) and modified for seawater by Breen et al. (2018). In this method, SCN is chemically modified with monobromobimane (MBB) to form a SCN-bimane product. The prepared blood plasma (50 µL) was added to a 1.7 mL microcentrifuge tube. A working solution of the internal standard (200 ppb NaS13C15N) (Cambridge Isotope Labs) and MBB (4mM) (Monobromobimane, Cayman Chemical 17097), in borate buffer (0.1 M, pH = 8), was prepared in a 2.5:1 ratio just prior to addition to the blood plasma. Then, this solution (35 µL) was added to each sample. The resulting solution was heated to 70 °C for 15 min to form the bimane-SCN complex. Standard solutions of SCN (5.0 ppb–25.0 ppm) were prepared in both HPLC grade water and in commercially available salmon blood plasma (MyBioSource inc., San Diego, CA, USA) and were derivatized in the same manner as the samples. As described in Breen et al. (2019), five-point calibration curves were run using standards in HPLC grade water prepared with MBB in the same manner as the samples just prior to all blood plasma analysis, such that the concentration of standards bracketed the expected blood plasma concentration.

The derivatized samples were analyzed using a UHPLC and column conditions described in Muñoz Muñoz et al. (2017) which comprised an Acquity UPLC® I-Class (Waters Corp., Milford, MA, USA) and Q-Exactive™ hybrid Quadrupole-Orbitrap™ high-resolution accurate mass (HRAM) mass spectrometer (Thermo Fisher Scientific, Waltham, MA, USA). A Kinetex 1.7 µm XB-C18 100 Å 2.1 ×50 mm column held at 40 °C was used for the gradient chromatographic separation (Phenomenex, Torrance, CA, USA) with 10 mM ammonium formate as mobile phase A and 10 mM ammonium formate in methanol as mobile phase B. The flow rate was 0.25 mL/min, and gradient conditions were as follows: 10–100% B (0.00–3.00 min), 100% B (3.00–4.00 min), followed by 2.00 min of re-equilibration time at initial conditions (total chromatographic run time 7.00 min). The Q-Exactive source conditions in electrospray ionization negative mode were as follows: sheath gas flow rate 55, auxiliary gas flow rate 15, sweep gas flow rate 2, spray voltage 4.50 kV, capillary temperature 300 °C, s-lens rf level 55, auxiliary gas heater temperature 500 °C. Tracefinder™3.2 (Thermo Fisher Scientific, Waltham, MA, USA) was used for data acquisition and processing. Thermo Xcalibur™3.0 (Thermo Fisher Scientific, Waltham, MA, USA) was also used for data processing.

The Q-Exactive acquisition method comprised a full-scan of 50–300 m/z with a resolving power of 70,000. Analyte identity was established relative to a standard by scoring the following qualitative criteria using TraceFinder 3.2: retention time (rt  ± 0.15 min), full-scan accurate mass (± 5 ppm window), and full-scan isotope pattern (scores range 0-100, and a score of ≥ 70 was used as the positive cutoff). Analyte quantitation was performed using the peak area ratio from the full-scan extracted ion chromatograms (XIC) of the analyte (248.04992) and its internal standard (250.05031); the XIC mass window was the accurate mass ± 5 ppm in all cases.

Analyte validation and quantification followed Breen et al. (2018), validation of peak area ratio (analyte/internal standard) and quantification by a linear weighted (1/x) regression of calibration standards. Plasma, water, or seawater calibration standards were prepared fresh for each run –no legacy calibration curves were used. Calibration standards were used only if the calculated concentration deviated ≤ 10% from the nominal concentration. The analyte response at the lower limits of quantification and detection (LLOQ = 5 ppb and LOD = 2.5 ppb) had a signal-to-noise ratio (S/N) ≥ ten and three respectively.

Statistics and analysis

The SCN half-lives and the accompanying regression statistics were determined as previously described by Breen et al. (2019) using Origin 2018 (OriginLab, Northampton, MA, USA). The data were fit using the exponential fitting tool to a single exponential decay function, (y=A1e−xt1+y0), where x is time, y is concentration, y0 is the value of the function at the asymptotic limit, A1 is the concentration maximum, and t1 is the reciprocal of the first-order elimination rate constant k. None of the variables were constrained and no weighting function was used. Fits were to the full data set, the blood plasma concentration for each fish measured was treated as an individual sample point and thus were not averaged at each sampling interval prior to fitting. The reported half-life is related to the rate constant by t12= ln2k .

When a two-compartment model was employed, a graph of the natural log of the SCN blood plasma concentration vs days post-exposure was constructed, and the data were divided by inspection into a fast or early time component and a slow or later time component (Breen et al., 2019). Each component group was fit to a linear function and the half-life was calculated from the resultant slope of the linear fit, t12= ln2slope.

Results

Cyanide exposure

Shortly after the fish were immersed in the 50 ppm CN bath, they began to swim erratically and gasped for air. While in the recovery bath, a loss of balance but strong respiratory activity was observed. Both 20 s and 45 s exposed fish were completely immobilized by 45 s after initially being dipped into the CN bath. For the 20 s exposure, fish returned to normal swimming behavior approximately 7 min after exposure while for the 45 s, normal swimming returned by approximately 17 min, indicating that recovery time varied proportionally with the exposure time. There were no mortalities in any exposure. Most exposed fish did not accept food for the first 3 days post-exposure, but then ate regularly after the third day.

After exposure to CN, the SCN concentration in the blood plasma was observed to increase quickly over the first 6 hrs, and then begin to decrease rapidly over the next 24 hrs. The maximum SCN was observed 4 hrs post-exposure corresponding to a concentration of 468 ± 29 ppb for the 45 s exposure and 3 hrs post-exposure corresponding to a concentration of 301 ± 6 ppb for the 20 s exposure (Figs. 1A, 1B). In this first CN exposure, SCN blood plasma levels began to plateau at day 3 and remained elevated until the last sampling time at day 13 (158 ± 54 ppb). Control fish SCN plasma levels for this trial were (112 ± 21 ppb, n = 5). To determine when SCN blood plasma levels reach control levels, a second 45 s CN exposure was carried out. In this second trial, the SCN blood plasma concentration peaked at close to 400 ppb over the first two time points sampled (4 and 13.2 hrs). Control levels for this exposure (15 ± 4 ppb, n = 4) were reached seven days after exposure (Fig. 1C). Between these two cyanide exposure experiments, additional control fish (n = 9) were sampled for their SCN blood plasma levels. Eight were found to be below detection limits and one was found to be 35 ppb.

Figure 1 The SCN blood plasma concentration during depuration in A. clarkii after exposure to 50 ppm CN.

Trial 1 (A) 20 s and (B) 45 s with sampling out to 13 days and trial 2 (C) 45 s with sampling out to 72 days. The black solid line represents the fit of the data to a single-phase exponential decay function. Gray data points represent early time points where SCN plasma levels are increasing and are therefore not included in the fit. Each data point indicates the mean ± S.E. The dashed gray lines represent the average SCN concentration in control fish for a given exposure. Where no error bar is observed the error is smaller than the data point. Inset is the natural log of SCN blood plasma concentration as a function of time and includes both trials of 45 s exposure data. These data were divided into two groups and a linear fit was performed on both the fast (blue) and slow (red) component of SCN elimination.

In all CN exposures, the rate of SCN elimination was fit to a single-phase exponential decay function with time constant parameters (Table 2). Regression statistics demonstrated all exposures to be statistically significant when fit to a single-phase exponential decay function (T1, 45 s: r2 = 0.55 and P < 0.0001, T1, 20 s: r2 = 0.68, P < 0.0001, T2, 45 s: r2 = 0.86, P < 0.0001). The data were fit without constraining y0, and thus the concentration of SCN was not forced to go to zero at infinite time. This resulted in y0 values for the first trial of 163 ppb ± 15 for 45 s exposure and 136 ppb ± 10 for 20 s exposures which are above control levels and 7.8 ppb  ± 17 for the second trial which is below control levels. In the first trial, the half-lives observed were fast, dropping to a plateau level in about two days following exposure. The plateau level remained for the next 13 days, and no data were obtained for times longer than 13 days for this trial. In the second trial, the control levels were reached quickly, no plateau was observed, and the resultant half-life was 1.2 ± 0.2 days. In comparing the goodness of the fit from trial one to trial two for the 45 s exposures, the trial two fit is much better than either of the trial one fits, based on the reported correlation coefficients and standard errors, however this could be due to the differences in the intervals in which sampling was conducted. For the first trial, in order to define the curve at early times and capture SCN levels going through a maximum, 8 times points were sampled in the first 3 days post-exposure. For the second 45 s trial, only 3 time points were sampled in the first 2 days. As with A. ocellaris, attempts to fit all the 45 s exposure data to two-phase exponential fit in Origin to use a two-compartment model did not converge. However, the longer half-life was determined to be 42 ± 33 days for the 45 s exposure using a two-compartment model (Fig. 1C insert). It is worth noting that there was an outlier for the four fish sampled at the last time point at 72 days post-exposure, but the point was not eliminated. The concentrations of SCN in the blood plasma for the four fish sampled at 72 days were found to be 12, 27, 33, and 120 ppb SCN. If the outlier (120 ppb) is removed, then the estimate of the half-life becomes 28 ± 16 days and the SCN blood plasma levels have returned to near control levels.

Table 2 The results of the fit of SCN blood plasma concentration versus time to the function y=A1e−xt1+y0 in order to determine the elimination half-life of SCN from the blood plasma of Amphiprion clarkii for both acute exposure to CN chronic exposure to SCN.

The standard errors reported with the fit results are also given. k and t1/2 are calculated from the fit parameter t1.

Parameter	45 s exposure, trial 1	20 s exposure, trial 1	45 s exposure, trial 2	SCN exposure trial 1	
t1/2 (days)	0.10 ± 0.04	0.20 ± 0.06	1.2 ± 0.2	0.13 ± 0.02	
Y0 (ppb)	163 ± 15	136 ± 10	7.8 ± 17	-580 ± 1700	
A1 (ppb)	880 ± 390	261 ± 54	468 ± 35	55000 ± 4000	
t1 (days)	0.14 ± 0.06	0.29 ± 0.09	1.7 ± 0.3	0.19 ± 0.03	
k (days−1)	6.9 ± 2.7	3.5 ± 1.0	0.58 ± 0.09	5.3 ± 0.8	
r2	0.55	0.683	0.86	0.90	
P	P < 0.0001	P < 0.0001	P < 0.0001	P < 0.0001	
n	47	44	41	37	

Thiocyanate exposure

Throughout the chronic SCN exposure (100 ppm, 12 days) experiments, fish behaved normally, ate well, and showed no external signs of stress. Once depuration was initiated, blood plasma levels were found to be at a maximum (44 ppm ± 2.5) at the initial sample time (30 min) and decreased to control levels by day 15 (Fig. 2). The detection of low levels of SCN in the plasma of A. clarkii using HPLC-UV was limited by the presence of a large neighboring peak not observed in salmon or A. ocellaris blood plasma. Fitting of the data to a single-phase exponential decay resulted in a half-life of 0.13 ± 0.02 days with an r2 value of 0.90 (Table 2). SCN levels in holding water of these chronically exposed fish were found to be in the 20–50 ppb range for the first few hours after depuration started. No SCN was detected after the first water change (24 hrs) even though sampling continued for 72 hrs.

Figure 2 The SCN blood plasma concentration during depuration for Amphiprion clarkii after exposure to 100 ppm SCN for 12 days.

The black solid line represents the fit of the data to a single-phase exponential decay function. Each data point indicates the mean ± S.E. Where no error bar is observed the error is smaller than the data point. Inset shows later time points.

Holding water

A second chronic exposure trial was undertaken to further examine the holding water result. Fish (n = 10) were exposed to SCN (100 ppm, 12 days) in a single 20 L polycarbonate tank, then held in individual beakers during the depuration period. The results of the holding water tested for SCN upon depuration from this second chronic exposure are shown in Fig. 3. A water sample from each beaker was collected immediately before fish were introduced to ensure no SCN was in the holding water (no-fish control). Water samples were also collected immediately after introducing a fish to its beaker (t = 0). In both instances, SCN was below our detection limit. At the next sampling point, 2 hrs after the start of depuration, SCN levels in the holding water were found to be at their highest level of 39 ppb ± 5. After this time point, SCN concentrations decreased continuously and, when sampled at 24 hrs but before the first water change, no SCN was detected in the holding water. Sampling continued just prior to daily water changes for two additional days but no SCN in the holding water was detected at these later time points. The SCN blood plasma levels were also measured for the fish used in this study. At 48 hrs, half the fish were sampled, and their SCN blood plasma concentration was found to be 74 ppb ± 27. The remaining half were sampled at the final time point, 72 hrs, and the SCN blood plasma was found to be 26 ppb ± 6.

Figure 3 The average (n = 10) concentration of SCN detected in 0.5 L of holding water containing a single Amphiprion clarkii depurating for 72 hrs after exposure to 100 ppm SCN for 12 days.

The data points indicate the mean ± S. E of the SCN concentration in the holding water. Circles are samples once depuration had begun; the square indicates the measured concentration of SCN in the holding water just prior to the addition of the fish. Where no error bar is observed the error is smaller than the data point.

This observation of a steady decrease of the SCN concentration in the holding water of a depurating fish over the first 24 hrs from an initial maximum required further investigation to see if indeed, the reduction of SCN from the holding water was real and repeatable. To test this, cultured fish with no known prior exposure to SCN were placed in beakers containing 500 mL of seawater that was spiked to 20 ppb SCN and sampled over 24 hrs. The spike concentration of 20 ppb was much lower than the 100 ppm used in the chronic exposure studies, but was on par with the concentration observed in the holding water from the earlier depuration study. When a single fish was held in 500 mL of seawater spiked to 20 ppb, the concentration of SCN measured in the holding water continually decreased over the first 24 hrs studied. Initially, the SCN concentration in the water was determined to be 15 ± 1 ppb but this dropped to below our detection limit at 16 hrs (Fig. 4) and remained below the detection limit until the “re-spike” after the first water change at 24 hrs. Here again, the SCN concentration sampled immediately after spiking was determined to be 15 ± 1 ppb but was below our detection limit (1 ppb) when next sampled 24 hrs later. SCN blood plasma levels were measured for the fish used in this experiment and found to be 429 ppb ± 164 and 429 ppb ± 94 at the 24 hr and 48 hr sampling times respectively.

Figure 4 The average (n = 10) concentration of SCN detected in 0.5 L of holding water spiked with SCN containing 1 non-exposed Amphiprion clarkii.

Each data point indicates the mean ± S. E. Where no error bar is observed the error is smaller than the data point. The round data points indicate the data collected after the initial SCN spike (t = 0). The square data points indicate the data after a water change at 24 hrs followed by a second spike.

Discussion

CN exposure

Amphiprion clarkii exhibited similar behavior to A. ocellaris when exposed to CN. Erratic behavior was followed by a loss of equilibrium and paralysis. In comparing these results with our previously reported work on A. ocellaris (Breen et al., 2019), the maximum level of SCN in blood plasma was decidedly lower for A. clarkii than for A. ocellaris while the half-lives reported were similar or smaller for A. clarkii depending on the exposure time (Table 3). The lower concentration of SCN in the blood plasma suggests that A. clarkii take up less CN during exposure when compared to A. ocellaris or eliminate metabolites from the blood plasma more efficiently. The half-lives reported here for acute pulsed exposure in marine fish are in reasonable agreement with those reported for mammals using intravenous or subcutaneous injections (0.21–8.0 days) (Logue et al., 2010).

Table 3 Summary of SCN half-lives in fish exposed to CN or SCN including relative standard deviation (RSD) and average fish weight.

Summary of max concentration of SCN observed in fish exposed to CN or SCN.

Model species		CN 20 s trial 1	CN 45 s trial 1	CN 45 s trial 2	SCN chronic	Reference	
Amphiprion ocellaris	Half-life (days)	0.44 ± 0.15	1.01 ± 0.26	NA	0.35 ± 0.07	Breen et al. (2019)	
Percent RSD	34%	26%	20%	
Mass(g)	5.68 ± 1.42	5.03 ± 1.41	3.84 ± 0.59	
Max. Conc (ppm)	1.9 ± 0.6	2.3 ± 0.2	220 ± 31	
Amphiprion clarkii	Half-life(days)	0.20 ± 0.06	0.10 ± 0.04	1.2 ± 0.2	0.13 ± 0.02	Current study	
Percent RSD	(30%)	(40%)	(17%)	(15%)	
Mass(g)	3.07 ± 0.62	3.54 ± 0.78	12.84 ± 9.37	6.09 ± 2.72	
Max. Conc (ppm)	0.301 ± 0.066	0.468 ± 0.028	0.399 ± 0.060	44 ± 2.5	
Oncorhynchus mykiss	Half-life(days)	NA			2.02 ± 0.06	Brown et al. (1995)	
RSD			3%	
Mass(g)			20	
Max. Conc (ppm)			60.5 ± 6.2	

In the first CN exposure trial, the measured half-life of the 45 s exposure appears faster than the 20 s exposure, contrary to those results for A. ocellaris. However, the large relative standard deviations (RSD) (40%, 30%) associated with these data make us hesitant to draw any dramatic conclusions. Our second 45 s trial, which had a much lower RSD than the first 45 s trial (17% vs. 40%), resulted in a half-life similar to that observed for A. ocellaris. The goal of the second trial was to establish when control levels were reached, and because of this, there were only 3 times sampled within the first three days of depuration. Trial 1 CN exposures were sampled at 8 time points over the first three days, and have a large variability, resulting in a less robust fit of the elimination from the blood plasma because of natural variability.

In the first trial of the current study, the elimination of SCN from the blood plasma plateaued at approximately 150 ppb starting at 4 and 12 hrs for each exposure time respectively. Likewise, in A. ocellaris, SCN concentrations in the blood plasma plateaued at approximately 500 ppb for both 20 and 45 s exposure times beginning at two and four days respectively (Breen et al., 2019). The lower plateau in this study is likely due to the shorter half-life and initial lower levels of SCN observed in their plasma.

Vaz et al. (2012) exposed A. clarkii (1.8 ± 0.2 g) to 25 ppm CN for 60 s and reported a 33% mortality rate. We did not observe any mortality when fish were exposed to CN concentrations of 50 ppm CN for 45 s. However, this difference in vulnerability could be due to differences in fish size, health and/or stress level (Hanawa et al., 1998). When assessing the vulnerability of A. ocellaris to CN, Madeira et al. (2020) found that larger fish had a higher survival rate and quicker recovery time when exposed for the same time and concentration of CN as their smaller conspecifics. As the fish used in our study were larger than those in the Vaz et al. (2012) study, a lower mortality rate would be expected.

SCN exposure

As with the CN exposure, when the results for A. clarkii were compared with those previously reported for A. ocellaris (Breen et al., 2019), the SCN blood plasma level was much lower. Both species were exposed to 100 ppm SCN bath for 11–12 days, but the maximum SCN blood plasma was 44 ppm ± 2.5 for A. clarkii, while that for A. ocellaris was at 220 ppm ± 31 (Table 3). The SCN levels in the blood plasma of A. clarkii were half that of the exposure bath, while those observed for A. ocellaris were twice that of the exposure bath. It appears that even under different exposure conditions (pulsed CN versus chronic SCN), A. clarkii have lower levels of SCN in their blood plasma, indicating that they uptake less CN/SCN during exposure than A. ocellaris or eliminate SCN from their blood more efficiently.

The half-life for SCN clearance in the blood plasma measured for A. clarkii when chronically exposed to SCN was more than two times shorter when compared to A. ocellaris. In a similar experiment on the freshwater fish rainbow trout (Oncorhynchus mykiss) the reported half-life for clearance of SCN from their blood was 2.02 or 2.36 days depending on the model used (Brown et al., 1995). This was much slower than our reported value of 0.13 ± 0.02 days and could reflect the differences in the osmoregulatory systems of marine versus freshwater fish.

The continued failure to replicate the work of Vaz et al. (2012) calls to question the ultimate fate of CN and SCN in marine fish (Breen et al., 2019; Breen et al., 2018). We have now confirmed in two marine species that CN is converted to SCN quickly as evidenced by the rapid rise of SCN in the blood plasma following CN exposure, which is also in accordance with the mammalian model. The clearance rate of SCN from the blood plasma is also fast, with the highest levels depleted within the first few days of CN exposure, also in accordance with mammalian models. However, the question remains: where does the SCN go? Despite multiple attempts, SCN was not detected in the holding water of fish acutely exposed to CN, but elevated levels of SCN in the blood plasma were observed after CN exposure (Breen et al., 2018; Breen et al., 2019). Is the failure to detect SCN in the holding water of marine fish post-acute exposure to CN because they are not excreting SCN, or is it because any excretion by a small fish in a relatively large quantity of water is diluted to concentrations below the detection limit?

Holding water

Blood plasma levels for A. clarkii of SCN following acute exposure to CN were found to be in the range of 500 ppb at their maximum but decreased rapidly after exposure. Levels above this are unlikely to be observed as higher CN doses would be required, leading to an increase in mortality rather than higher levels of SCN in the blood plasma (Breen et al., 2019; Madeira et al., 2020). In order to increase SCN blood plasma levels and thus enhance the likelihood of detecting SCN in the holding water of a marine fish, A. clarkii were exposed to 100 ppm SCN for 12 days and allowed to depurate in only 500 mL of holding water. Under such conditions, blood plasma levels were initially found to be close to 50 ppm for A. clarkii, approximately 100 times higher than the highest blood plasma SCN concentration observed in fish exposed to CN. Under these optimized conditions for recovery, SCN was detected in the holding water of these chronically exposed, depurating fish. The observed levels of SCN in the holding water were 1000 times less than SCN blood plasma levels in chronically exposed A. clarkii (Fig. 2) and 2000 times less than the SCN concentration fish were exposed to. Most surprisingly, SCN holding water levels sharply decreased from 40 ppb observed at 2 hrs to undetectable levels within the first 24 hrs of depuration. Sampling continued before each water change for two additional days, but no further SCN was detected in the holding water even though SCN blood plasma levels were found to be 74 ppb ± 27 at 48 hrs and 26 ppb ± 6 at 72 hrs for the depurating fish. The decrease in SCN concentration in the holding water of A. clarkii while depurating demonstrates that A. clarkii have the capability of removing low levels of SCN from their holding water and do not continually release SCN to their holding water, even when SCN remains elevated in their blood plasma.

While the source of the SCN detected 2 hrs after introducing the fish is unknown, once the SCN is introduced to the holding water, it is likely re-absorbed into the fish as it is no longer present in the water. Breen et al. (2018) ruled out container effects and demonstrated that SCN concentrations were stable in control beakers without fish. It was also observed that SCN concentrations were significantly reduced with beakers containing one, A. ocellaris (Breen et al., 2018). The most likely explanation for this uptake of SCN is either via diffusion across the gills or through the intestines via the drinking response. However, drinking rates in marine fish that have been reported range from 2–7 mL/kg per hour making it implausible for the fish to drink 500 mL of holding water in less than 24 hrs leaving the gills as the likely site of entry (Fuentes & Eddy, 1997; Grosell, 2019; Perrott et al., 1992). This study was not designed to determine the mechanism of the removal of SCN from holding water by fish, and further work is required to determine the removal mechanism(s).

Our results of holding water spiked with trace levels of SCN demonstrate that with a fish present, the SCN concentration decreases to below detectable levels within the 24 hr period after the spike. This removal of SCN from the holding water is accompanied by elevated levels of SCN in the blood plasma. Our holding water results are in marked contrast to Vaz et al. (2012) who claimed that fish pulse-exposed to 25 ppm CN for 60 s excreted SCN starting at day 2 and that excretion continued even with daily water changes through their 28 day observation period without ever decreasing. They reported an initial increase and then a plateau of SCN in holding water at 9.84 ± 0.03 ppb. In our study, SCN was detected in the holding water for only the chronically exposed A. clarkii likely due to the higher SCN blood plasma levels (100×) than when CN exposed. The measured SCN in the holding water was only observed for the first few hours once depuration started, as this is when the SCN in the blood plasma is at its maximum and we again note that the SCN concentration in the holding water was 1000× less than the measured SCN in the blood plasma during these early times. As detailed by Breen et al. (2018), the amount of SCN that Vaz et al. (2012) claimed their CN exposed fish excreted is too high and we again reject the notion that SCN can be detected in the aquaria water of CN exposed fish, not at Day 2 nor at Day 28.

It has often been erroneously suggested that cyanide metabolites are excreted out of the fish via urine following CN exposure (Vaz et al., 2012) as in mammalian models (Bruckner & Roberts, 2008; Lanno & Dixon, 1996; Logue et al., 2010; Nelson, 2006). It is widely accepted that the mechanism of uptake and elimination of ions from marine teleost depends on the charge of the ion. Uptake of monovalent anions occurs from diffusion across the gills and transport in the intestines. Excretion of monovalent anions is accomplished by active transport via chloride cells in the gills (Greenwell, Sherrill & Clayton, 2003). Divalent ions enter via the drinking response but are excreted in the urine (SO42−, Mg2+) or fecal matter (Ca2+) (Evans, 2008; Grosell, 2019; Hickman Jr, 1968; Whittamore, 2012). There is no evidence that the urine is the primary pathway to eliminate monovalent anions in marine teleost (Whittamore, 2012).

This difference in SCN elimination compared to mammalian models is due to the osmoregulation strategy of marine fish. To maintain osmotic balance with the surrounding water, marine fish have a low urinary excretion rate in order to retain water in their blood. The urine flow rates in marine teleosts are 1–2% of body weight daily because water is highly conserved and can be reabsorbed by the urinary bladder (Evans, 1993). However, while we demonstrate that SCN is rapidly eliminated from blood plasma, the terminal fate of SCN in the body of marine fish remains unknown but must be consistent with the mechanisms of monovalent ions.

The data presented here demonstrate that marine fish can remove trace quantities of SCN from the holding water and retain it. This further corroborates the findings that testing for SCN in the holding water of this fish is not a viable indicator of CN exposure (Breen et al., 2019; Breen et al., 2018; Herz et al., 2016), opposing previous studies (Vaz, Esteves & Calado, 2017; Vaz et al., 2012). This also nullifies any concern of false positives of non-exposed fish during cohabitation with CN exposed fish.

The results of the A. clarkii SCN half-life following CN exposure suggest a two-compartment model for SCN elimination similar to that of A. ocellaris in that a fast and slow elimination component was observed (Breen et al., 2019; Toutain & Bousquet-Mélou, 2004). Two-compartment models are commonly used in pharmaceutical research for drug metabolism (Ishida et al., 2016; Metzler, 1971). In both of these species of marine fish, CN is absorbed through the gills and/or the gut via the drinking response and is quickly metabolized into SCN as evidenced by the rise in SCN blood plasma levels. We speculate that the tissue may be acting as a SCN reservoir. If this is the case, then SCN stored in the tissue of fish could potentially serve as an indicator of CN exposure but further studies quantifying SCN levels in the tissue of marine fish exposed to CN would need to be conducted.

In order to properly validate a test for CN exposure by the presence of SCN, it is also critical to know the baseline values of SCN in the blood and tissue of non-cyanide caught fish from the areas where CN is likely used. Fish in the Indo Pacific region may be exposed to very low concentrations of CN or SCN from natural sources, such as cyanogenic foods, foods rich in SCN, or anthropogenic sources, predominantly runoff from industrial processes such as mining. Therefore, these fish may already have significant SCN levels in their blood and/or tissues. The present findings confirm the blood plasma SCN maybe be a useful biomarker of CN exposure in marine fish if measured shortly after exposure. Additional species from a broader taxonomic sample must be evaluated prior to any definitive conclusions.

Supplemental Information

Supplemental Information 1 Raw Data

Compiled master data file supporting the present investigation. Each tab is representative of a particular experiment. The instrument generated data files are also available in the Supplemental Files.

Click here for additional data file.

Supplemental Information 2 Mass spec data from DD summarized in CN 1 45 s plasma tab and CN 1 20 s plasma tab in compiled master data file

Click here for additional data file.

Supplemental Information 3 Reinjection of mass spec data from DD summarized in CN 1 45 s plasma tab and CN 1 20 s plasma tab in compiled master data file

Click here for additional data file.

Supplemental Information 4 Mass spec data from DD summarized in CN 2 45 s plasma tab in compiled master data file, fish 335-352

Click here for additional data file.

Supplemental Information 5 Mass spec data from DD summarized in CN 2 45 s plasma tab i in compiled master data file, fish 353-358

Click here for additional data file.

Supplemental Information 6 Mass spec data from DD summarized in CN 2 45 s plasma tab in compiled master data file, fish 359-366

Click here for additional data file.

Supplemental Information 7 Mass spec data from DD summarized in CN 2 45 s plasma tab in compiled master data file, fish 367-371

Click here for additional data file.

Supplemental Information 8 Mass spec data from DD summarized in CN 2 45 s plasma tab in compiled master data file, fish 372-375

Click here for additional data file.

Supplemental Information 9 Mass spec data from DD part 1 summarized in SCN 12 d Depuration Water tab in compiled master data file

Click here for additional data file.

Supplemental Information 10 Mass spec data from DD part 2 summarized in SCN 12 d Depuration Water tab in compiled master data file

Click here for additional data file.

Supplemental Information 11 Mass spec data from DD part 3 summarized in SCN 12 d Depuration Water tab in compiled master data file

Click here for additional data file.

Supplemental Information 12 Mass spec data from DD summarized in SCN 12 d depuration plasma tab in compiled master data file

Click here for additional data file.

Supplemental Information 13 Mass spec data from DD summarized in SCN spike Water tab in compiled master data file

Click here for additional data file.

Supplemental Information 14 Mass spec data from DD summarized in SCN spike plasma tab in compiled master data file

Click here for additional data file.

Supplemental Information 15 Peak Simple Chromatograms for Chronic SCN Exposure

Click here for additional data file.

We would like to thank the current and former undergraduates from Roger Williams University who have been involved in this project, especially Julia Grossman, Gabbie Baillargeon, Hannah Sterling, Natalie Danek, Julia Dwyer, and Sara Hunt. Kevin Erickson of MASNA provided thoughtful comments on an earlier draft of this manuscript. We also thank the reviewers for their helpful comments and insight. Sea & Reef Aquaculture of Maine U.S.A. provided a portion of the fish used in these experiments.

Additional Information and Declarations

Competing Interests

Author Contributions

Animal Ethics

Data Availability

Lawrence J. Andrade is an employee of Dominion Diagnostics.

J. Alexander Bonanno, Nancy E. Breen and Andrew L. Rhyne conceived and designed the experiments, performed the experiments, analyzed the data, prepared figures and/or tables, authored or reviewed drafts of the paper, and approved the final draft.

Michael F. Tlusty conceived and designed the experiments, analyzed the data, authored or reviewed drafts of the paper, and approved the final draft.

Lawrence Andrade analyzed the data, authored or reviewed drafts of the paper, and approved the final draft.

The following information was supplied relating to ethical approvals (i.e., approving body and any reference numbers):

Roger Williams University Institute of Animal Use and Care Committee approved this research (R180820).

The following information was supplied regarding data availability:

The raw data is available in the Supplemental Files.

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
