# Peer review of "The determination of thiocyanate in the blood plasma and holding water of Amphiprion clarkii after exposure to cyanide"

_PeerJ, doi:10.7717/peerj.12409_

## Round 0.1 · original submission · Major Revisions

I now have three detailed reviews of this manuscript, each of which recommended significant revisions for clarity and tone. As this paper addresses an ongoing debate about the outcomes of fish exposure to cyanide, it is particularly important that the reviewer comments be addressed in a thorough and thoughtful manner. I would ask the authors to submit their revised ms. along with a point-by-point response to each of the reviewers' comments, addressing, in particular, concerns about methods, statistical analyses, and limitations of the conclusions based on the results. The same reviewers will be asked to evaluate the revised manuscript.

In the interests of full transparency, the following text was provided by the editorial staff regarding the review of this paper:

"CRITIQUE PAPER: Please note that this work critiques work published by the laboratory of Ricardo Calado. It is PeerJ policy to invite the lead author of a paper to comment as a reviewer when we receive such a critique and so we would ask that you invite him (rjcalado@ua.pt) and/or Diana Madeira (d.madeira@ua.pt) as well as at least two additional reviewers. They will be required to sign their review in the interests of transparency and we would strongly encourage the authors to publish the review history if the manuscript is accepted."

Reviewer 1 ·

Basic reporting

This is an excellent study in an area in need of advancement. The overall research is excellent but some of the English could be improved. My comments are below:

1) The early part of the introduction has multiple areas where it is difficult to understand what is being referred to. For example, the use of "This is critical" at the beginning of a sentence refers to the last sentence which has multiple concepts. Therefore, readers may be confused as to what "this" refers to. I have suggested some minor edits in the attached PDF file for clarity.

2) The statement on Line 107 is a very speculative statement since perhaps at some point there will be a method capable of detecting very low levels of SCN excreted from some fish. I know you found SCN is actually absorbed by the Clarkkii and not excreted, but that may not apply to all fish. Also, you can presume the reader has yet to see the SCN in holding seawater results that you present later in the paper. Essentially, you shouldn't say something is impossible unless it defies the laws of physics. I think you can just go on to say that aside from testing SCN in holding water, many groups have suggested analysis of SCN in the blood of CN exposed fish. Even as far back as the NOAA panel on CN detection.

3) On line 111, was 41 days post exposure the last day tested. If so, you should denote this.

4) Line 129, the abbreviation UPLC is associated with a specific brand. The common name for it is UHPLC. Also, this abbreviation should be spelled out.

Experimental design

Very good experimental design, very similar to a previously peer-reviewed and published article. My concerns on the treatment of the data are mainly:

1) Line 142. "housing fish in a temperature-controlled room" sounds like the fish are walking around in the room. You need more detail about how they are housed.

2) Lines 160-166. It would be better to have this info below in the plasma extraction section. Alternatively, you could or bring the plasma extraction section up into this section. If doing that, you should remove the plasma extraction section. Please either totally separate the exposure from the sampling and analysis (as implied by the current headers) or bring everything into one section. As it is now, it reads as if there is missing information at first, and then when I got to the next sections, some of that info is redundant and I got confused as to what was new info and what was info that was repeated.

3) Line 170. "rinsed thoroughly by the treatment group" is confusing.

4) Line 172. "in the same groups as exposure" is confusing.

5) Lines 173-175. Same as comment 2.

6) Lines 178-179. When exactly were the water samples collected? Was it immediately? If there was a set time (e.g., “2-min prior to water changes"), then please add.

7) Lines 186-188. This is confusing. I’m thinking that you bled half the fish at 24 hr and the other half at 48 hr by the way it is written, but it is unclear.

8) Need "x g" (g-force) along with, or instead of, rpm for all centrifuge speeds. Lines 200 and 204 are examples. For all centrifuging, temperatures should also be given since cold centrifuging is common.

9) The plasma precipitation section is not significant enough to be a whole separate section. Add to the SCN analysis section.

10) Line 213. Don’t get this statement when the sample was dried. Unless it isn’t dried completely. I think you just need to give the volume of water you added to reconstitute.

11) Starting Line 233, there are multiple mistakes in the units. The worst one is using millimeters (mm) to represent millimolar (mM).

12) Line 242. This is called "resolving power" and calculated as full width at half maximum (FWHM).

Validity of the findings

The discussion has significant gaps. My concerns are below:

1) Line 274. There were only two times, so this statement is very speculative. You can perhaps just compare the recovery of the 20 s exposure fish to the 45 s exposure fish. That they took approximately double(???) the amount of time to fully recover. Also, you may want to caveat this if you didn’t accurately measure this by saying something like, while not specifically recorded for each fish, we observed that recovery time for the 45 s exposure was approximately double(???)….

2) Figure 1. Did you ever look at a two-compartment model? Basically you could plot the log SCN concentration vs time and see if there are two distinct linear phases or only one straight line. If there are two, it is likely a two-compartment model instead of a 1-compartment model. A 2-compartment box model has a fast phase and then a slower phase, similar to what Breen 2019 described. If you have a two-compartment model, then there would be two t1/2s and the second one would be longer. That would account for why the SCN doesn’t return to control levels for the 15 days in A and B. Figure C looks like a one-compartment model, which I can’t really wrap my head around. Why do A and B have high consistent levels of SCN for 14 days and probably longer, whereas C goes to the control pretty quickly (at about 5 days), but then pops back up above control at 72 days. This should be discussed. I didn't see much discussion about this. I see in the discussion section that you do briefly discuss a two-compartment model, but it seems that you did not try to actually fit the data using this model to see if it is more appropriate than the one-compartment model used. The paragraph starting on line 461 states that the data suggest a two-compartment model, but the fit of the data was assuming a one-box model.

3) Figure 1 caption. For reporting values with error, you should report them as 34 ± 10 ppb. Also, why use the ">" symbol. The line represents one value.

4) Line 284. Why does it say 31 ppb here (Line 307 also) and 34 ppb in the figure caption?

5) Please abbreviate hours and minutes in the paper.

6) Line 303. Do you want to say something about trial 3 data fits here?

7) Figure 2. Since this is in ppm and Figure 1 was in ppb, there should be an inset showing the ppb scale as a comparison to the figure above. This would give readers a sense as to if the SCN falls back to baseline very quickly or if it stays above the control line like in Figure 1 A and B above. That would give us a sense as to if the elevated levels of SCN after CN exposure are due just to the SCN depuration or if they have something to do with the SCN coming from CN (e.g., does CN get trapped or bound to something and slowly convert to SCN, producing the higher than background levels of SCN for days).

8) Lines 311-313. The wording implies you were not careful before. Also, it doesn’t really give the reader a sense as to what is different or why we have this section.

9) Line 320. I don’t understand. Why is this positive? What was the purpose of the first experiment? Do you need to even discuss the first experiment? If this was a preliminary experiment just to see what would happen, maybe only the detailed experiment needs to be presented. It might make the results much more clear.

10) Line 323. Are these 3 different times? It seems like it. Or is it one time point. If so, it is written in a confusing way. If there are 3 different times, commas should be used to make this clear or even stating that there were 3 sampling times.

11) Figure 3. Were any of the points in Figure 3 below the LLOQ of the method? If so, they should not be plotted since they cannot be confidently quantified. Another option is plotting <LLOQ concentrations as zero, but then this should be discussed in the caption. Additionally, I don’t think you need the legend since you discuss the shapes in the caption. If you want the legend, it would be best to have a more descriptive title than “pre-fish” and the circles are post-addition of fish and should not be labeled SCN concentration since all points are SCN concentrations.

12) Line 329 and the whole section. This section is very difficult to understand. Were you changing the water? After much time, my understanding is that the SCN came out of the fish and then went back in and changed to something else maybe? I don’t know if I’m looking at this right, but it took me a really long time to understand (if I even do) this section. I think you need to stress how you did these experiments, maybe with a “timeline” figure for all the experiments.

13) Line 330. It would be good to be specific as to why it required further study.

14) Figure 4 caption. What does the "20" spike mean?

15) Line 348. A faster half-life doesn’t necessarily suggest that the clarkii take up less CN during exposure. It has more to do with the metabolism of the clarkii vs the ocellaris.

16) Line 353. Can you do a t-test here to have some statistical measure to back up this claim?

17) Line 358. Seems like an argument for a 2-box model, even though you fit with a 1-box.

18) Line 421. You kind of contradict yourself and then just leave the reader hanging here.

19) Line 437. Did you spike SCN into the containers with no fish and determine if the concentration was constant? If you don’t have that control, you cannot say that SCN is not sorbed by the walls of the vessel. Think this control is a must and the results of this experiment should be discussed.

20) Line 442. You can't say the fish retain SCN. SCN that is taken up may be metabolized to something else.

21) Line 452. Seems redundant with previous sentence. Combine?

22) Starting Line 472. The logical flow seems to break down a bit here, jumping from point to point without transitioning. Maybe this paragraph could be integrated into another one?

23) The missing piece in the discussion is…Where does the SCN go? The title of the paper asks the question and it isn't resolved. It certainly is eliminated from the blood plasma, but doesn't seem to go into the holding water (assuming controls show it doesn't get sorbed to the container). In mammals, SCN is eliminated via urine. In fish, it must be used in some metabolic process or is stored somewhere. I think storage of SCN (as SCN) is quite unlikely, but I don’t know the biochem of marine fish well. If someone could figure out where it actually ends up (storage in the fish or conversion to a metabolite), there might be an alternative marker of CN exposure that is excreted into the holding water, which could be used for detection of CN exposure.

Additional comments

General comments are listed in the other sections.

Annotated reviews are not available for download in order to protect the identity of reviewers who chose to remain anonymous.

·

Basic reporting

The document does not require major English editing but, in its present form, it requires several clarifications detailed bellow in the section “General comments for the author”.
References need to be carefully revised, as some of the in-text citations are not listed, some listed references are incomplete, and others are incorrect.
There are several claims in the introduction that reflect the authors opinion and not scientific facts. These must be revised accordingly.
The structure of the Methods section needs a major revision, as in its current version it is extremely hard to follow. It is not easy to understand the number of trials performed (or their rationale) and the number of replicates employed per each sampling point (including that of control fish). A scheme would be most helpful.
Raw data have been made available but some supplementary information is still needed.

Experimental design

The work does not clearly define a research question.
While the authors state how their study may contribute to fill current gaps of knowledge, they assume some premises that are yet to be demonstrated for marine fish (see Lines 132-134). The build-up and excretion of SCN- originating from the detoxification of an acute pulse-exposure to CN- may be distinct from that when this same compound is absorbed by a fish through a chronic exposure. The methods section is too confusing and needs clarification.

Validity of the findings

The data on which the conclusions are based are provided by the authors and made available as a supplementary excel file, but more information is needed on the methods employed (such as calibration curves). Replication is modest and the number of replicates (including control fish) per sampling point needs to be clarified. This issue must be clearly discussed.
The conclusions presented fail to discuss some relevant findings by the authors, such as the prevalence of SCN- in blood plasma 72 days post-exposure, as well as why have these levels increased above those recorded at day 18 and 50 post-exposure. The authors also do not discuss the basal concentration of SCN- in the blood plasma of their control fish. The interpretation of the findings reported by the authors is much supported on their claims on their publication Breen et al. (2018), based on their assumptions that other studies are not valid.

Additional comments

Title
The title of a manuscript is supposed to be informative. This is not the case in this publication, as the authors fail to answer the question “Where does it go?”. With their experimental set-up the authors are unable to determine “the fate” of thiocyanate in the blood plasma of a tropical reef fish that has endured an acute pulse-exposure to cyanide. As such the title must be revised to better reflect the findings reported (such as the remarkably long-term persistence of thiocyanate in the blood plasma following this type of exposure).


Abstract
Lines 23-25
This is not a fact supported by any scientific reference, as to date no study confirmed that SCN- is indeed the major metabolite originating from CN- exposure in marine fish. All relevant sources of scientific literature address terrestrial vertebrates, namely rat, pig, rabbit, and man. Moreover, four processes are involved in toxicokinetics: absorption (substance enters the organism), distribution (substance moves from the site of entry to other areas in the organism), biotransformation (the organism transforms the substance in new metabolites) and excretion (substance or its metabolites leave the body). These four processes are not detailed in this study and, as such, the term toxicokinetics should not be used.

Line 30 and along the whole manuscript
Do not express post-exposure times as decimals of days. Expressing it in hours is much more informative to the reader.

Line 31 and along the whole manuscript
Please say “blood plasma” instead of simply “plasma”.

Lines 36-38
How can the authors demonstrate that a fish is unable to excrete a certain compound by showing that the fish is able to uptake it? The rationale supporting this claim certainly needs clarification using a solid scientific support to allow this claim to stand, namely when the authors refer that it is “refuting several publications”. Please clarify.

Lines 38-40
However, the authors do not try to discuss if SCN will remain in the fish tissues indefinitely, nor if any alternative pathway of excretion (other than urine) occurs. For how long will SCN remain in fish tissues? Indeed, “Where does it go?”, the question in the manuscript title that the authors fail to answer. The presence of SCN in the blood plasma of fish pulse exposed to an acute dosage of CN 72 days post-exposure is not discussed by the authors. This is a very important finding that should be addressed in the abstract.

Lines 40-41
This claim by the authors is rather surprising as the authors refer that SCN is distributed throughout the fish tissue, is not excreted in their urine and they have been able to detect it in the fish blood plasma up to 72 days (!) post-exposure to an acute dosage of CN. If SCN is not excreted, why will it not be possible to detect it in fish tissues? Please elaborate.
May the long-term persistence of SCN in the blood plasma of fish once exposed to cyanide poisoning (up to 72 days!) eventually be a caveat to formulate a causative relation between the presence of this metabolite and cyanide poisoning? Please elaborate.

Introduction
References need to be carefully revised, as some in text citations are not listed, some listed references are incomplete, and others are incorrect. Only as an example (Line 72), the work by Cervino et al. (2003) only addresses the effect of cyanide exposure on hermatypic corals and anemones, not fish. The claim being attributed to these authors is in fact supported by the citation of Rubec (1986) and Rubec and Soundararajan (1991) in the work by Cervino and collaborators. Please delete this reference and cite, if you will, the original works (after checking them). Another example (Line 85), if the reference Rubec et al. 2008 refers to the book chapter listed in the reference list, the correct year of the publication is 2003. Additionally, this reference does not support what the authors refer on the manuscript being revised as it says: “The IMA adopted and implemented the standard operating method for the determination of total cyanide ion published by the American Society of Testing and Materials (ASTM 1997), and the American Public Health Association (APHA 1998). The test procedure has undergone extensive evaluation by U.S. government agencies. Likewise, the IMA conducted quality assurance/quality control procedures that demonstrated that the CDT procedure was reliable (Manipula et al. 2001b)”. Please see page 330 of the book chapter. Correct the sentence or delete the reference.

Line 87-88
“The test required lethal sampling…”, so does the approach proposed by the authors to survey SCN concentrations in the blood plasma of marine fish (at least the small sized ones that dominate the trade of marine ornamental reef fish). The authors do not address this issue in their discussion, and they should. They must discuss how the use of a test that requires sacrificing fish may be caveat to the implementation of any testing. This is a major bottleneck when surveying live fish whose intrinsic value for either the marine aquarium trade, or premium restaurants in some southeast Asian countries, is the fact of them being live. How will the industry handle routine testing that confirms that the hundred- or thousand-dollars fish sacrificed was clean of any metabolic trace of cyanide poisoning? Will more fish be collected from coral reefs to compensate those that may eventually be killed during screening? Please address this issue in the discussion.

Lines 89-91
There is no text in Logue et al. (2010) referring to fish that says “Any test for CN caught fish must be administered very soon, likely within hours, after exposure”. Can the authors please indicate where in this reference Logue and co-authors refer this?

Lines 92-93
For accuracy, the sentence should say “As an alternative to detecting CN directly, thiocyanate (SCN), the major metabolite of CN exposure in terrestrial vertebrates, can be used as an indicator of CN exposure (Youso et al. 2012)”.

Lines 96-98
Not once in the paper by Vaz et al. (2012) is the word "mammals" referred. This claim is an interpretation by the authors of the manuscript being revised. Vaz et al. (2012) only hypothesize that SCN- is excreted through the fish urine without making any analogy to mammals, nor discussing how similar (or dissimilar) these excretion pathways can be between marine fish and mammals. The sentence must be corrected or deleted.

Line 98
“In a now refuted study, Vaz et al. (2012)…”
This claimed refutation of Vaz et al. (2012) is based on an oversimplified interpretation of the cyanide kinetics in marine fish pulse-exposed to an acute dosage of this toxicant and relies on data from three marine fish published in 1981 in an aquarium magazine (which was not peer reviewed). This study performed almost 40 years ago was clearly framed by its author, Prof. David R. Bellwood, as follows: “This experiment is far from conclusive”. The words “In a now refuted study” must either be deleted or replaced by “In a now challenged study”.

Lines 101-103
To date, the sole laboratory that has tried to replicate the screening for SCN- in the water used to depurate tropical reef fish pulse-exposed to an acute dosage of cyanide is that of the authors of the work by Breen et al. (2018) (three of the them co-author the present submission being revised: Nancy E Breen, Lawrence J Andrade and Andrew L Rhyne).
The study by Murray et al. (2020) is a bibliographic review with no experimental component. Hence, Murray et al. (2020) did not try to replicate or refute the methodology described by Vaz et al. (2012). Murray et al. (2020) solely refers to Breen et al. (2018) questioning the findings by Vaz et al. (2012). As such, Murray et al. (2020) should not be cited in this sentence.
Concerning the study by Herz et al. (2016), it is puzzling that even after being informed that all analysis of this publication were performed not in Germany by the authors of that work, but rather in Portugal by the authors of Vaz et al. (2017), the corresponding author of the study being revised still cites Herz et al. (2016) as a source support the claim being made. The chromatograms obtained from the samples of Herz et al. (2016) were not indicative of the absence of SCN from the water being analyzed, as they displayed a "bulge like curve" that prevented the identification of any well-defined peak. Consequently, it was impossible to know whether SCN was present in these samples or not. The work by Herz et al. (2016) was sent to publication without the knowledge, or the consent, of any of those involved on the laboratory analysis of the samples (which are the core of that study). The authors of Vaz et al. (2017) contacted the editor of SPC Live Reef Fish Information Bulletin reporting this procedure by the authors of Herz et al. (2016). The editor acknowledge that he had not been informed on this issue and that the study had not been peer-reviewed prior publication. In this way, for the sake of transparency Herz et al. (2016) should not be cited either.
The above being said, only the reference by Breen et al. 2018 should be used, and for the sake of transparency, the words "even though multiple labs have attempted to do so (Herz et al. 2016; Breen et al. 2018; Murray et al. 2020)." should be replaced by "even though Breen et al. (2018) have attempted to do so.".

Lines 103-106
The claims by Breen et al (2018) result from an oversimplified approach to cyanide toxicokinetics in marine fish. To the authors best knowledge, these claims are not supported by at least one peer-reviewed publication specifically addressing this issue in a marine fish.
The concentration of CN entering marine fish at which the enzymes involved in detoxification pathways are no longer able to impair cytotoxic hypoxia is species specific, depends on fish biomass and can be influenced by temperature (see Madeira et al. 2020 https://doi.org/10.3389/fmars.2020.00246). How much CN can enter a marine fish prior to the saturation of detoxification pathways, how quickly CN is fully converted into less toxic metabolites and what are the maximum levels of SCN that can accumulate in marine fish during a sub-lethal pulse-exposure to CN are yet to be clarified. Again, to date, and to the authors best knowledge, no peer-reviewed scientific publication has ever reported these values. It is also unknown in what organs of marine fish is SCN preferentially stored prior to its excretion.
The interpretation of Breen et al. (2018) cannot simply disregard CN toxicokinetics in marine fish, neither CN strong and variable affinity to living organic tissues, or potential interactions with other molecules). The efficiency of detoxification pathways (how quickly and for how long do these pathways allow the fish to endure an exposure to CN- by converting this ion to SCN-, Vitamin B12 or ATCA) cannot be overlooked. If this over-simplification was scientifically acceptable the development of physiologically-based pharmacokinetic (PBPK) models and the development of computer-based predictions of biochemical constants, usually based on great amounts of data from laboratorial exposures, would not be necessary for the risk assessment of chemicals to humans and other animal species.
The striking differences in excretion mechanisms recognized for freshwater and marine fish species does not allow one to assume that these organisms display similar toxicokinetics on what concerns the detoxification of CN and doing so can be a major pitfall. The reader is not made aware of this potential caveat in the work by Breen et al. (2018), where data from Ramzy (2014) referring to Nile Tilapia, a freshwater fish, are presented to support the fact of Vaz et al. (2012) data being “unrealistic”. Moreover, as referred by Ramzy (2014), only CN stability was measured in that study, with no analysis being performed to quantify the presence or persistence of any cyanide-derived metabolites (e.g., SCN-) originating from detoxification pathways.
Concerning data retrieved from Bellwood (1981) published in a popular aquarium magazine (hence not peer reviewed) cited in the work being revised and Breen et al. (2018), the reader is informed that this was a preliminary study that targeted solely three specimens of the violet damselfish Neopomacentrus violascens. The following sentences are direct quotes from Bellwood (1981): “There are many problems interpreting these results.”, “Specific tissues within an organ may also concentrate the cyanide, resulting in effects different from those expected from an analysis of the whole organ. It is also likely that the distribution will change considerably with time.” and “This experiment is far from conclusive.”. The cautionary approach by Bellwood (1981) to his very preliminary trials are not considered by Breen et al. (2018), as they fail to inform the reader that from the three fish tested by Bellwood (1981), two had been starved for 24 h and one had been fed a heavy meal prior to CN exposure (which impacts the entrance of CN through their digestive system); there is no information on the time of exposure other than that fish remained in the cyanide solution for an additional 30 s post-anesthesia; and the muscle of the, which approximately represents 50% of the whole biomass, was not screened in the study (this may bias results given the strong affinity of CN to well blood irrigated tissues, such as muscle, which is also one of marine fish tissues with higher levels of rhodanese).
For the reasons detailed above it is not legitimate that the authors of the present publication to say that “a mass balance calculation demonstrated that the SCN levels reported by Vaz et al. (2012) were not possible”.

Line 107
This claim does not reflect a fact, as detailed in the comment above on the so-called mass balance study referred by Breen et al. (2018). For the sake of transparency, the sentence should be rephrased as follows: “With the aquarium water test being disputed, testing for SCN in bodily fluids and tissues of exposed marine fish is the next logical step.”

Line 113
Replace “much slower elimination rate” by “much slower elimination rate from the blood”, as in the way it is written it may misguide the reader to assume the authors are referring to elimination from the fish, which is not the case. Please correct.

Lines 114-115
“Breen et al. (2019) speculated that the observation of both a fast and slow elimination rate might be due to multiple elimination pathways in marine fish”.
What do the authors mean by elimination pathways? Do they truly mean elimination, and not retention? What they advocate in the present publication is that SCN resulting from CN detoxification is retained within fish tissues. They refute urinary excretion so what alternative pathways are the authors referring to? Please clarify and support your claims with peer reviewed references.

Line 118
Replace “plasma” by “blood plasma”, here and along the whole manuscript.

Line 124
Replace “endogenous levels of SCN in marine fish” by “endogenous levels of SCN in marine fish in the wild”.

Lines 129-130
As referred above, the four processes of toxicokinetics were not covered in this study.

Lines 132-134
The authors assume that SCN entering a marine fish through a chronic exposure will behave the same as when it originates from the detoxification of CN. How do the authors support this assumption? Are there any scientific references to support this assumption?

Materials and Methods
This section is too confusing, only when reading the Results can one understand what trials were performed and for what purpose. The section needs to be rearranged and clarified. Some schematic representations summarizing the trials performed (detailing sampling times in hours and days, not as decimals of days), the conditions tested, if trials were an acute pulse-exposure or a chronic exposure, the number of fish on sampling time and in the control (not only the total n) would be much helpful.

Line 145
Correct the values for temperatures (and other measurements along the manuscript) making sure you add a space between the value and the unit, so it reads XX ºC; salinity has no units (delete here and elsewhere in the text), simply refer 30. Why was a salinity of 30, and not 35, used in this study? Any relevant reason?

Line 147
Exposure
For clarity, always refer to the trials being described/discussed as acute pulse-exposure or chronic exposure. Please correct

Lines 148-149
What is the rational supporting the use of 20 and 45 seconds as exposure times? The lack of standardization of experimental trials increases the challenges of comparing results.

Lines 152-153
Were all fish stocked together or was there one fish per 20-L tank? If fish were not stocked individually, the experimental unit is the stocking tank and not the fish. The fish from the same stocking tank are pseudo-replicates and not true experimental replicates. Please clarify. Was natural seawater used? How was it filtered? Was this water monitored to make sure it was SCN free? It is known that coastal waters may display variable levels of SCN. Please detail these issues related with water filtration and control.

Lines 158-159
Please indicate how many control fish were monitored per each post exposure time being monitored. This is not detailed. Control fish should have been surveyed at the beginning of the trial (to determine the baseline of SCN in their blood plasma), as well as on each post-exposure time being monitored.

Line 190
Plasma extraction
Please provide at least one reference supporting these procedures.

Lines 192-195
Can the use of tricaine methanesulfonate interfere with cyanide metabolism? Any information on this? The authors must at least raise awareness on this issue in their discussion, as the ecotoxicological study of mixes raises data interpretation to an extra level.

Line 208
Plasma precipitation. Please cite at least one reference supporting this procedure. Were the periods of centrifugation different from samples of small and big fish? Please detail.

Line 216
Thiocyanate analysis
What was the derivatization protocol employed to generate SCN-bimane? Please provide a reference.
Please detail if all blood plasma and seawater samples were analyzed using UPLC-MS. Additionally, if SCN concentration in all samples was verified using UPLC-MS, why is there a reference to analysis also being performed using HPLC-UV? This will likely only confuse the reader. Please clarify this issue.

Lines 217
Bhandari et al. (2014) is missing from the reference list. On what do the methods of Breen et al. (2019) differ from those used by Bhandari et al. (2014), so both references are cited?

Lines 226-228
Where the calibration curves prepared with the internal standard (200 ppb NaS13C15N) employed to monitor concentration using UPLC-MS?
Where the calibration curves prepared with the derivatized (SCN-bimane) standards?
The calibration curves employed to analyze blood plasma and seawater should be made available, at least as a supplementary file.

Line 229
When the authors say samples, are they referring to both blood plasma and seawater samples. Please clarify the text.

Line 244
“total chromatographic run time 7.00 min”
Given the total chromatographic run time employed it is important to make available at least some chromatograms and mass spectra for blood plasma samples and certainly for seawater exhibiting SCN as supplementary material. Please provide these.

Line 265
The authors refer that “Fits were to the full data set…” but on the caption of Figure 1 they also refer that “early time points where SCN plasma levels are increasing and are therefore not included in the fit”. Please clarify this and discuss the significance of the procedure employed of not including early time points on the curve being adjusted.

Lines 283-288
The rationale for the second acute pulse-exposure to CN performed should be framed in the Materials and Methods, not here.

Lines 290-291
There is no reference to these analysis on the subsection “Statistics and analysis” in the Materials and Methods. Please correct.

Lines 302-303
On the caption of Figure 1 the authors refer that “early time points where SCN plasma levels are increasing and are therefore not included in the fit”. Please also clarify this issue.

Line 303
Before advancing to the next paragraph presenting the results on the chronic exposure to SCN, the authors must refer that by day 72 post-exposure to an acute pulse-exposure to CN, the levels of SCN recorded in the blood plasma of fish were well above those of days 18 and 50 and no longer in line with the baseline levels recorded for control fish. This is an important finding and must be clearly stated in this section and interpreted (in the Discussion).

Line 307
Why is there no baseline of SCN levels in blood plasma provided for control fish in Figure 2 as there is for Figure 1? Please detail.

Line 313
For accuracy, instead of referring “multiple fish” please detail the number of fish used.

Lines 317-318
Where are these control checks detailed in the Materials and Methods and what methodology was used to perform them (HPLC-UV or UPLC-MS)? Please detail.

Lines 319-321
The rationale for the trials performed must be outlined in the Materials and methods.

Lines 330-333
Again, this information must be framed in the Materials and methods.

Discussion
The authors do not discuss the relevance of their findings reporting SCN being detected in blood plasma 72(!) days post-exposure, well above the baseline determined for control organisms, as well as above the levels recorded from conspecifics 18 and 50 days post-exposure to an acute pulse-exposure to CN.
The authors also fail to discuss that this type of screening is lethal for the fish being surveyed. They should discuss how this issue may cause resistance to the testing being discussed from the live fish trading industry. This caveat may even be more relevant for pricier fish, some of them already flagged in previous studies as potentially being targeted by cyanide fishing.

Lines 349-350
What the authors have determined in the preset study are SCN half-lives in blood-plasma, not CN half-lives. The values from Logue et al. (2010) summarized in Table 1 for SCN half-lives range from 4.95 to 192 hours (not 0.21-8.3 days). Please clarify this issue and interpret your findings framed with the contrasting cyanide exposure methods employed for marine fish (a bath, with CN entering through the fish gills and mouth) and mammals (subcutaneous or intravenous injections).

Lines 354-355
The explanation provided refers that there were fewer early time points sampled. But were these early time points used or excluded from the fit (as reported in the caption of Figure 1)?

Line 360
For clarity, replace “SCN concentration for A. ocellaris” by “SCN concentration in the blood plasma for A. ocellaris” as the reader maybe misled to believe the authors are referring to the whole fish, which is not the case.

Line 362
For the same reason as above, replace “the SCN concentration” by “the SCN concentration in blood plasma”.

Line 370-371
As the size range of specimens used in the second acute pulse-exposure trial to CN was high (according to Table 1 the largest specimen was four times bigger than the smallest one) may this have been a source of error? Please discuss this issue.

Lines 375-377
Please frame this data with the respective times post-exposure at which the fish were screened. This information is important to frame this comparison.

Lines 384-388
Why is it relevant to the authors to make this comparison with a freshwater fish? As they refer, the osmoregulation of marine and freshwater fish is so strikingly different that providing examples from freshwater fish only adds noise to an already noisy discussion on CN kinetics in marine fish. These sentences, as well as data for O. mykiss from Table 3, should be deleted.

Line 396-399
It may also be because other detoxification metabolites are being produced, or not? Please elaborate.

Lines 400-401
The authors must frame this with the post-exposure time at which the first fish enduring an acute pulse-exposure were screened. If the first record of SCN in blood plasma refers to 1 h post-exposure and 7 or 17 minutes post -exposure (for an acute pulse exposure of 20 s and 45 s, respectively) the fish were already recovered and swimming, how high may SCN concentrations be in the periods between the end of the pulse exposure and this first record? The authors must discuss this and frame their data within this context.

Lines 401-403
This is a assumption, not a fact. It would all depend on how quickly and for how long the CN detoxification machinery could handle the CN dosage. To date, no data on this key aspect is available for marine fish.

Lines 416-417
Why should the entrance pathway of SCN to the fish being chronically exposed to it be through its drinking response? Can the authors provide any scientific supporting this assumption?

Lines 417-420
Why are the authors excluding urinary excretion? The only likely explanation is that by admitting that urinary excretion of SCN may indeed exist, their main claim that questions the validity of the work by Vaz et al. (2012) needs to be reconsidered. Is there any scientific support for this hypothetical retention of SCN in the slime of fish? Were these fish also rinsed post their chronic exposure, as the ones pulse-exposed to CN, to avoid any confounding effects? The authors must also acknowledge that the SCN being considered results from a chronic 10-days exposure and not from CN detoxification. The modes of action displayed by the fish to handle SCN originating from different sources (one exogenous, the chronic exposure, the other endogenous, CN detoxification) may also be different. To date, no one knows.

Lines 421-424
If this hypothesis is unlikely, why is it presented? It seems it is a way for the authors to avoid admitting that maybe fish can excrete SCN through urinary excretion?

Lines 425-427
How may have SCN been excreted? Through urinary excretion? If so, are marine fish able to excrete SCN through their urine? Is this what the authors are saying here? Please elaborate.

Lines 431-434
For accuracy, the authors should refer that this finding refers to fish that were pulse-exposed to 25 mg/L of CN. In the other trial performed by Vaz et al. (2012) using a pulse-exposure to cyanide of 12.5 mg/L SCN was only detected 6 days post-exposure, not 24 h.

Lines 439-440
Again, this is the authors interpretation, not the claim by Vaz et al. (2012), as already referred above.

Lines 440-442
But on lines 426-427 the authors refer "it is also possible that most of the SCN absorbed in the chronic exposure may have been rapidly excreted." So, are marine fish able or not able to excrete SCN? The authors are illusive on their claim on this ability by marine fish and never refer ipsis verbis "marine fish are not able to excrete SCN". Please clarify.

Line 444
Concerning the reference to Herz et al. 2016, please see the comment referring to Lines 101-103.

Line 445
Vaz et al. 2017
The authors fail to acknowledge that Vaz et al. 2017 deals with marine fish captured from the wild, not pulse-exposed to an acute dosage of CN in the laboratory, and that the values of SCN recorded in the holding water of these fish are in the same order of magnitude as those reported earlier by Vaz et al. (2012). The corresponding author of the present study is also aware that, although performed in the same university and department, the researchers involved, the laboratories and equipment used in Vaz et al. (2012) and Vaz et al. (2017) were not the same (which validates the methodologies employed and the data retrieved). All chromatograms of Vaz et al. 2017 were also made available as supplementary material for that publication (as per request of one of the anonymous peer reviewers that revised the study). No subsequent publications on this topic (Breen et al. 2018 and 2019) have disclosed the chromatograms of their experiments, neither does the present study being revised. Please clarify this issue or delete Vaz et al. 2017 from the sentence.

Lines 445-446
How did the findings by the authors nullify this concern, if non exposed fish stocked in water spiked with SCN absorbed this compound? The concentration of SCN used to spike the water (15 ug/L) is clearly within the range of SCN reported by Vaz et al. 2017 screening fish collected from the wild, as well as the values recorded in some coastal waters reported in scientific literature. If fish unexposed to CN are stocked in natural seawater (instead of artificial seawater as advocated by Vaz et al. 2012 and 2017), they may indeed absorb SCN from the water. As referred by Madeira and Calado (2019) (https://doi.org/10.1016/j.ecolind.2019.03.054), it is urgent to standardize experimental methodologies on this topic to advance the state of the art. The use of artificial seawater should be preferred over natural seawater to enhance the control over experiments and replicability.

Lines 453-455
The rates presented by the authors are valid for fish under homeostasis. What happens when fish are exposed to a toxicant? Is urinary excretion affected? The release of SCN from the slime coat of the fish is somehow far-fetched and likely only advocated by the authors not to refer that SCN may indeed be eliminated by marine fish through urinary excretion. Please provide some references that may support your hypothesis of SCN being released from the slime coat of fish.

Lines 459-460
Exactly! No one knows what happens to SCN- ions and the authors fail to answer the question in their title. The authors refute the urinary excretion pathway for the elimination of SCN- but refrain from saying ipsis verbis "marine fish are not able to excrete SCN". Why?

Lines 463-465
So where do the authors suggest that CN is detoxified into less toxic metabolites in marine fish? Are the authors suggesting that this process only occurs in the gills? This is not likely, as rhodanase has been reported to be active in many other organs, such as muscle and most importantly, the liver. This build-up of SCN in the plasma results from SCN being released from the tissues where it was produced from CN detoxification? If so, why is it then sequestered in the tissues and removed from the blood plasma? The authors should elaborate on this.

Lines 470-471
If this is so, and it may well be, for how long will SCN stay in the fish? Forever? Any thoughts on this by the authors? According to the data being reported, fish may display SCN in their blood plasma above the baseline values of conspecifics non-exposed to cyanide poisoning up to 72 days post-exposure! This may be both an advantage and a caveat for any test aiming to reveal the illegal use of cyanide poisoning to collect live marine fish. The authors should discuss this finding and highlight the advantages and disadvantages of such a long-term persistence of this metabolite.

Lines 478-480
What cyanogenic foods are the authors aware in the marine realm? Cyanogenesis is not as common in the marine environment as in terrestrial habitats and this claim certainly needs some scientific support. Please elaborate.

References
Please carefully revise each reference.

Figures
Figure 1
To facilitate the "stand alone" reading of each figure, please include near the letter of each panel of the figure the pulse-exposure time (in s) and dosage of CN employed for the acute pulse-exposure performed.
Replace “plasma” by blood plasma.
In the statistical analysis subsection the authors refer "Fits were to the full data set, the plasma concentration for each fish measured was treated as an individual sample point and were not averaged at each sampling interval prior to fitting". Which one is valid? The information in the caption or in the Material and methods? If early data post exposure were left out because they do not fit the curve, is the exponential decay function selected adequate?
Do the authors have any explanation for the baseline values of SCN present in the blood plasma of their control fish? How many fish were screened and used as control for each of the trials presented and at which sampling times? Are these baseline values “normal”? Is there any prior reference to this in the literature?
Why are error bars so small in some of the data points?

Figure 2
Why is no baseline displayed for the levels of SCN on the blood plasma of control fish?
Replace “after exposure” by “after a chronic exposure”

Table 1
Replace “Average weight ( + 1 S.D.) and sample size (n) of Amphiprion clarkii exposed to 50 ppm CN for either 20 or 45 seconds across 2 trials and 100 ppm for 12 day.” by “Average weight ( + 1 S.D.) and sample size (n) of Amphiprion clarkii pulse-exposed to an acute dosage of 50 ppm of CN for either 20 or 45 seconds across 2 trials and a chronic exposure of 100 ppm of SCN for 12 days.”
Is there no effect of fish size on SCN half-life? Any statistics to support this?
Please also indicate the number of fish (n) per sampling time post-exposure and not only the total n; also indicate how many control fish were monitored and indicate the number of sampling times post-exposure monitored (as hours and days, not as decimals of days). Place the Table in landscape if necessary.
As for trial CN2, please indicate maximum and minimum weight (the authors can easily retrieve this information from their supplementary tables).

Reviewer 3 ·

Basic reporting

The abstract exceeds the 300 word limit and would need to be slightly more concise (by 47 words)
Line 96: remove “the” from “SCN is eliminated in the urine by..”
Line 97: Bruckner and Roberts (2008), referenced in the manuscript, similarly reports such speculation at the CDT workshop and may also be referenced here alongside Vaz et al. (2012).
Line 100 -101: sentence starting “The authors..”: It is noteworthy here that this approach was applied to monitor the marine aquarium trade in the EU by the same group (Vaz et al, 2017, referenced)
Line 102: replace “labs” with “laboratories”
Line 127: “the same species used by Vaz et al. (2012)”: is there any particular reason for highlighting this work here? If so, clearly indicate the purpose for doing so
Line 129: “UPLC-MS” and “HPLC-UV”: This is the first time these abbreviations are used in the manuscript and need to be spelled out. Please note that “UPLC” is a trade mark registered by the Waters Corporation.
Line 275: “into CN bath” add “the” between into and CN
Line 421: replace “observed” with “detected”
Line 440: reference to Vaz et al., 2012: Bruckner and Roberts (2008) report the same speculation and should be referenced here.
Line 447: replace “Howerver” with “However”
Line 459: sentence starting “The divalent..” remove “The”

Experimental design

Line 148 and 160: Provide the full company detail for Sigma e.g. Sigma-Aldrich, St Louis, MO, USA.
Line 212: “the acetonitrile was evaporated with warm nitrogen gas”: provide temperature if different to ambient and controlled.
Line 212 -213 “Auto Vap (Zymark)”: Auto Vap is not a system supplied by Zymark. Please provide full name of the system and company details (e.g. Zymark, Hopkinton, MA, USA).
Line 217-218: Suggest to clarify by adding “where SCN is chemically modified with monobromobimane (MBB) to form a SCN-bimane product.”
Line 221-223: sentence starting “To minimize pipetting errors…”: This sentence would benefit from clarification e.g. Larger volumes of the internal standard and the MBB solution were mixed at the appropriate ratio just prior to addition to the plasma.
Line 224-225: “Standard solution of SCN (10.0 ppb- 25.0 ppm)” was NaS13C15N (200 ppb) also added?
Line 225: “HPLC grade water and in commercially available salmon plasma”: please specify which calibration standards where use for the analysis of plasma/aquaria water.
Line 226: Sentence starting “ Five point calibration curves were prepared…” please indicate that the calibration curves were prepared by derivatisation with MBB as per sample preparation.
Line 233: “10mm” change unit to “mM”
Line 234: “10mm” change unit to “mM”
Line 238: Replace “esi” with “electrospray ionization”
Line 240: Tracefinder™ is a Thermo Fisher Scientific product
Line 241: Xcalibur™ is a Thermo Fisher Scientific product
Line 249-257: can the authors provide quality control/quality assurance details. Was the method validated for each matrix, were control samples (e.g. spiked plasma) analysed alongside samples to ensure day to day fitness for purpose, what were the LLOQ and LODs.
Line 251-252: Sentence starting ”To accept a calibration standard..”: this sentence reads like a guideline, rather than a quality statement. Please indicate whether the calculated concentration did deviate by less than 10%
Line 252 – 254: Sentence starting “ The analyte response…”: this sentence reads like a guideline, rather than a quality statement. Please provide LLOQ and LOD
Line 256: Sentence starting “Samples were..”: This sentence is ambiguous, were results confirmed? Also, why not use the UPLC-MS method straight away?
Line 287: add “45 s” after “a second” for clarity.

Validity of the findings

Line 354 -355: “the argument could be made that there is no difference within their respective uncertainties”: can the statistical significance be tested for robustness of this statement?
Line 446 -447: sentence starting “This also nullifies…” generalises to “fish” and should just refer to A. clarkii until more corroborating evidence is produced.
Line 463 – 464: sentence starting “Two-compartment…(Metzler, 1971)” The reference is dated, is this statement still valid in the present tense?
Line 479 – 481: sentence starting “Fish in these areas…” there are also significant anthropogenic sources of CN in these areas, such as mining or other industrial point sources that should be mentioned here.

Table 1: For row 3 “CN 1 45s”, n=48 but the raw data file supplied (peerj-48457-Bonannoetal_PeerJ2020_AClarkii_RawData) only list 47 fish. Also the average weight for this same exposure is given as 3.54g but works out as 3.56g in the raw data file.

Additional comments

This is an important contribution to the field of cyanide fishing as it further demonstrates that a post-capture test based on the measurement of SCN in aquaria waters, a once compelling prospect, is unlikely actionable. Instead, the evidence presented points to SCN being retained in the fish species studied. This leads the authors to question the fate of SCN in tissues of marine fish and its potential as a longer post-exposure indication of exposure to cyanide whilst reminding the reader that endogenous levels of SCN in fish pre-capture must be understood.

---

## Round 0.2 · Minor Revisions

Reviewers 1 and 2 have returned their comments on the revised ms. and both ask the authors to consider additional minor revisions. I wish to echo Reviewer 1 in congratulating the authors for providing a very thorough response to the first set of reviews. At this point, I will leave it to the authors to decide how they want to further revise their manuscript based on the second set of reviews.

Reviewer 1 ·

Basic reporting

No comment.

Experimental design

No comment.

Validity of the findings

The validity of the findings appear to be quite strong.

Additional comments

I wish to commend the authors on the most thorough responses I have ever seen. I have a few minor corrections which would hopefully improve the paper if implemented, but even if not implemented, I would recommend to accept the manuscript for publication.

1. Make sure it is very clear that you are utilizing the results of the Breen et al 2018 work as your control which proves that SCN isn't being lost to the container walls or some other mechanism. Be sure that you emphasize that you utilized the same experimental including the same type of holding containers.

2. Figure 1. The 75 day points on the inset...the top one seems different. Why? Also, maybe clarify that B and C are two different studies. I firstly thought it was two representations of the same data.

3. Figure 2. I think it would be clearer if you plot all the data in the main figure with breaks and then clearly show that you are zooming into the 0.75-18 days or so. My suggestion obviously implies that I think you should expand the zoom to those days not currently covered in the inset.

4. Figure 4. I think it would be clearer if you put arrows at the spike times labeling them as Spike 1 and Spike 2. Just the visual of the plot currently is highly unusual and will likely cause confusion.

·

Basic reporting

The document does not require major English editing but even on its revised form it requires several clarifications detailed bellow in the section “General comments for the author”.
References listed still need revision (e.g., Barber and Pratt 1997 has no source, neither has Day et al. 2018 or Losada and Bersuder; some references miss page numbers, such as Buno and Selig 2007). Please carefully revise all references.
The structure of the Methods section has been significantly improved but the number of fish sampled at each time point, particularly those of the control, must be clearly detailed. As already suggested in the first round of peer review, a schematic representation of the trials preformed signaling the number of exposed and control fish sampled at each time point (or at least a table clearly summarizing this) would make the life of the reader much easier.
Raw data have been made available, but some supplementary information can still be easily provided by the authors without requiring any major effort from their side. It is not easy to understand why the authors are so reluctant to provide some chromatograms and mass spectra for blood plasma samples, as well as for seawater exhibiting SCN, as supplementary material, given the total chromatographic run time reported. As referred by the authors, simply uploading as supplementary file the reports they have would do.

Experimental design

The description of the experimental design was significantly improved by the authors. However, for clarity, the authors need to detail how many fish were sampled as control on each sampling point of their exposure trials (a schematic representation would be most helpful to the reader). The number of fish sampled as control must be presented unequivocally and the reduced number of replicates of exposed fish sampled at each data point must be acknowledge. Although we do understand that welfare guidelines must be obeyed, these cannot compromise experimental designs due to low replication.

Validity of the findings

As previously referred in the first round of peer review, replication is modest and the number of replicates (including control fish) per sampling point needs to be clarified. The authors did not do so and refer that “It is against best animal care and use practices to overuse animals.” No one questions this ethical stand. However, this cannot be used as an argument for not having sound replication. As such, the authors must clearly refer how many fish were screened as control on each data point sampled in their exposure trials.

Additional comments

I must start by clearly referring that never have I, nor any of my co-authors, ever agreed to retract our publication “Vaz MC, Rocha-Santos TA, Rocha RJ, Lopes I, Pereira R, Duarte AC, Rubec PJ, and Calado R. 2012. Excreted thiocyanate detects live reef fishes illegally collected using cyanide—a non-invasive and non-destructive testing approach. PLOS ONE 7(4): e35355.”! What I told Dr. Rhyne in our last Skype talk, in a very unequivocal way, and after I realized that he was not just publishing an article saying that he could not replicate our study but rather pointing a finger saying that our data was fraudulent, was that I would sit down with my co-authors, go through all the steps of sample collection, analysis, and data processing (which after more than 6 years elapsing at the time was no easy task…) and if we had any reasonable doubt on the validity of our findings we would take action. The “selective amnesia” of Dr. Rhyne impaired him to refer in his letter that all samples of the above-mentioned publication were processed as blind samples (including blanks). How could it be possible that our data was fraudulent? We have carefully revised our study and found no reason, whatsoever, to take any action. However, Dr. Rhyne continued to claim our data were fraudulent clinging to a so called “mass model” that completely ignores the toxicokinetics of cyanide. Dr. Rhyne’s is yet to be able to answer the key question before anyone working on this topic can even start talking about toxicokinetics: how much cyanide enters a marine fish after a sub-lethal pulse exposure? Until this question can be answered, all research efforts on this topic have limited use, as only after knowing this may one try to understand if any concentration values being reported for the metabolites resulting from cyanide detoxification make sense (subsequently, variables such as duration of exposure, exposure dosage, fish size and species will also need to be investigated; only after knowing this should we start looking to the fate of cyanide that entered the fish, namely how it is metabolized and excreted).
Dr. Rhyne rushes to conclusions on a topic that all researchers to date still largely unknow. His position on this issue has drifted from one publication to the next, from “thiocyanate is excreted from pulse exposed fish in a matter of hours post-pulse exposure” to “thiocyanate is not excreted at all”, along with ”no such thing as cyanide derived metabolites will be stored for long periods after pulse exposure” to the most interesting finding reported in the present manuscript by him: the prevalence of thiocyanate post-exposure in the blood plasma of fish for, at least, 72 days post-pulse exposure. This finding is not even referred in the Abstract and is one of the most relevant findings of the present study!
Dr. Rhyne refers on his reply that “In order to move the science forward in this field, the reviewer should acknowledge the impossibility of Vaz et al. 2012 and retract the paper”. Why is this publication impairing Dr. Rhyne (or any other researchers) to pursue alternative hypothesis to the one presented on our publication? Dr. Rhyne was unable to replicate our method, then another method must be developed (by him or any other researchers working on this topic). This is what Dr. Rhyne promised to deliver to the funders of his research. To date, we are yet to see this breakthrough. When Dr. Rhyne compares the screening of seafood for antibiotics and that of live marine ornamental fish, he optimistically assumes that the industry will willingly accept sacrificing specimens worth hundreds of euros to receive a “the fish is clean” answer. Yes, clean but dead, with no use for the marine aquarium trade (a caveat that does not apply to the seafood trade, as most specimens are not traded live…). Moreover, such testing will not certainly be put in place in the EU (one of the largest importing markets for marine aquarium fish), as animal welfare principles will not tolerate this approach (sacrificing healthy specimens just to see if they have been exposed to cyanide). Putting the burden of testing fish on the side of exporting countries (as most are developing nations with limited financial resources) also fails to be a suitable approach if one truly aims to change the status quo.
Concerning the “biased” peer-review, I would just refer that both I and several of the co-authors of the publication being disputed serve on several editorial boards and revise tens of papers per year. Our previous review, as well as the present one, is as objective and constructive as possible. Dr. Rhyne cannot just question everything and everyone and each time a legitimate question is raised label it as a “red herring”. Neither I, or any of my co-authors, are supported by any NGO or any industrial partner with an interest in either shutting down the marine aquarium trade or trying to keep “business as usual” (respectively). Our interest in this topic remains purely academic.
For the sake of transparency, I sincerely hope that Dr. Rhyne makes available to any reader the full peer-review history of this publication (including that of the original submission).
Concerning specific comments on the revised study (please note that line numbers refer to the word version supplied with track changes with view as Simple Markup:
Dr. Rhyne and colleagues have improved the overall quality of the manuscript.
However, some issues still require attention by the authors before this work can be accepted for publication.
Abstract
One of the main findings of this study is not referred in the Abstract. The prevalence of SCN in blood plasma 72 days post a pulse exposure to CN. This must be referred in the abstract.

Introduction
Lines 86-90: As previously referred in the first round of peer review, to the authors best knowledge, the only laboratory that ever tried to replicate the method described by Vaz et al. 2012 was that of Dr. Rhyne. Again, the “selective amnesia” of Dr. Rhyne impaired him to recall that all samples of study Herz et al. (2016) (a non-peer reviewed publication whose “story” was previously detailed) were processed by Dr. Vaz in Portugal (not in Germany or Indonesia). While Dr. Rhyne insists on citing this study to support this claim, it is clearly written in the Acknowledgements of this publication “The authors wish to thank Dr Ricardo Calado, Dr Valdemar Esteves and Marcela M.C. Vaz from the University of Aveiro, Portugal, for analyzing the samples and for providing insights into their HPLC methodology. Their help is highly appreciated.”. Unfortunately, Herz et al. only “forgot” to inform those who performed the analysis that they were publishing this study… Murray et al. 2020 is a review study with no experimental work, whatsoever. As such, to practice the intellectual honesty that Dr. Rhyne so much advocated on his reply to our comments, either he writes “… it has never been replicated (as reviewed by Murray et al. 2020)” or says “... it has never been replicated (Breen et al. 2018).”. While on its reply to our comments it is referred that Murray et al. 2020 is referred as “reviewed by”, this correction was not performed. Inflating the number of references supporting this statement, as it is on its present form, is not acceptable.

Lines 90: Again, for the sake of intellectual honesty, where it reads “demonstrated by using a mass balance calculation” it should read “demonstrated by using a mass balance calculation (that did not consider the toxicokinetics of CN uptake and metabolization)”. It is not acceptable that every comment we make is a “red herring”, when we are asking for accuracy on the claims being made. If the method employed by Dr. Rhyne to perform the “mass balance calculation” could be used without knowing toxicokinetics, the life of ecotoxicologists working on risk assessment would be so much easier.

Lines 96-98: Studying the metabolites produced from the detoxification of CN in marine fish is of limited use until the key question is answered: how much CN enters a marine fish pulse exposed to CN. This is the first step of toxicokinetics. The authors should at least say something about this.

Methods
Line 136: Add a space so it reads “25 ºC”. Please revise throughout the manuscript.

Line 150: Please make it clear that only in the first trial were the fish pulse exposed to 20 and 45 s.

Line 152: Please clearly refer that in the second trial fish were only pulse exposed during 45 s in this second trial.

Line 155: Please clearly say how many fish were used as a control on each of the two trials. For clarity, please also refer how many replicate fish were sampled on each day post-exposure. As previously suggested, a schematic representation detailing the number of replicates of control and exposed fish sampled at each time point would be helpful for the reader.

Results
Line 307-308: Do the authors have any idea why one of their control fish displayed levels of SCN in blood plasma above the detection limit?

Line 309: This is confusing o the reader, as in the M&M section “CN and SCN Exposures” (as well as bellow in this paragraph) only two exposure trails are referred. A first trial with a duration of 13 days and two exposure times (20 and 45 s) and a second trial with a duration of 72 days and a single exposure time (45 s). I suggest the authors simply delete the word “three“ to avoid confusing.

Discussion
Lines 417-419: No need to cite the same reference twice in the same sentence. Just keep the citation at the end of the sentence.

Lines 439-442: There is another hypothesis that the authors cannot exclude, the possibility of other metabolites being produced to detoxify CN. As previously referred to Dr. Rhyne, available literature using mammals reports that, at times, the production of other metabolites may surpass that of SCN. The authors should at the very least refer this hypothesis, although no one has ever confirmed what other metabolites apart from SCN are produced by marine fish during the detoxification of CN. Please see Bhandari et al. 2014 Cyanide Toxicokinetics: The Behavior of Cyanide, Thiocyanate and 2-Amino-2-Thiazoline-4-Carboxylic Acid in Multiple Animal Models. Journal of Analytical Toxicology 38: 218 –225. doi:10.1093/jat/bku020

Lines 445-449: How does the reference Madeira et al. 2020 supports, this claim by the authors? The word “blood” or “plasma” is not mentioned a single time in that publication.

References
Several references listed are still incomplete (e.g., Barber and Pratt, 1997; Day et al., 2018). Please correct.

---

## Round 0.3 · accepted · Accept

The review process for this paper has been particularly thorough and the authors have done a good job of addressing the comments of the reviewers. It appears that the authors will be opting to make the review documents part of the record for this publication, which is a distinct advantage of publishing in PeerJ. Congratulations on an important contribution!